

# Modeling rhizosphere carbon and nitrogen cycling in *Eucalyptus* plantation soil

Rafael V. Valadares[1], Júlio C. L. Neves[1], Maurício D. Costa[2], Philip J. Smethurst[3], Luiz A. Peternelli[4], Guilherme L. Jesus[5], Reinaldo B. Cantarutti[1], Ivo R. da Silva[1]

[1]Department of Soils, Universidade Federal de Viçosa, Viçosa, 36570-900, Brazil
[4]Department of Microbiology, Universidade Federal de Viçosa, Viçosa, 36570-900, Brazil
[2]Commonwealth Scientific and Industrial Research Organisation, Hobart, 7005, Australia
[3]Department of Statistic, Universidade Federal de Viçosa, Viçosa, 36570-900, Brazil
[5]Celulose Nipo Brasileira S/A, Belo Oriente, Minas Gerais, Brazil,

*Correspondence to*: Rafael V. Valadares (rafaelvvaladares@hotmail.com)

**Abstract.** Vigorous *Eucalyptus* plantations produce $10^5$ to $10^6$ km ha$^{-1}$ of fine roots that probably increase carbon (C) and nitrogen (N) cycling in rhizosphere soil. However, the quantitative importance of rhizosphere priming is still unknown for most ecosystems, for instance *Eucalyptus* plantations. Therefore, the objective of this work was to propose and evaluate a mechanistic model for predicting rhizospheric C and N cycling in *Eucalyptus* plantations. The potential importance of the priming effect was estimated for a typical *Eucalyptus* plantation in Brazil. The process-based model (ForPRAN - Forest plantation rhizospheric available nitrogen) predicts the change in rhizosphere C and N cycling resulting from root growth and consists of two modules: (1) fine root growth, and (2) C and N rhizosphere cycling. The model describes a series of soil biological processes: root growth, rhizodeposition, microbial uptake, enzymatic synthesis, depolymerization of soil organic matter, respiration, mineralization, immobilization, microbial death, microbial emigration and immigration, SOM formation. Model performance was satisfactory quantitatively and qualitatively when compared to observed data in the literature. The input variables that most influenced N gain by rhizospheric mineralization were (in order of decreasing importance): root diameter > rhizosphere thickness > soil temperature > clay content. The priming effect in a typical *Eucalyptus* plantation producing 42 m³/ha/year of wood, with assumed losses of 40 % of the total N mineralized, was estimated to be 24.6 % of plantation N demand (shoot + roots + litter). The rhizosphere cycling model should be considered for adaptation to other forestry and agricultural production models where the inclusion of such processes offer the potential for improved model performance.

## 1 Introduction

Nitrogen (N) is a nutrient essential for plant growth and sustainability of natural and managed ecosystems, including *Eucalyptus* plantations (Barros and Novais, 1990; Jesus et al., 2012; Pulito et al., 2015; Smethurst et al., 2015). Despite high demand by the trees, low N availability commonly limits plantation growth, and plantations on soils with low organic matter concentrations are the most severely affected (Barros and Novais, 1990; Pulito et al., 2015; Smethurst et al., 2015). This is



attributed to the fact that most of the N absorbed by trees comes from decomposition of soil organic matter, i.e. N mineralization (Barros and Novais, 1990; Pulito et al., 2015; Smethurst et al., 2015).

Measurements of *in situ* net N mineralization are laborious, but can be predicted to some degree using models. Smethurst et al. (2015) evaluated a process-based model (SNAP) for estimating net N mineralization in *Eucalyptus* plantations in southeastern Brazil. The authors estimated annual rates of net N mineralization ranging from 148 to 340 kg/ha per year of N in the 0-20 cm soil depth, with additional available N expected in deeper soil layers. These rates of N supply were similar to or high than the N demand of young plantations in the region, and therefore consistent with the observation that plantation growth responses to N fertilization were minor or absent. An extension of the *in situ* core measurement used can estimate N uptake by plantations and has been independently validated (Smethurst and Nambiar, 1989). However, spatial and other methodological errors in this core technique are high. One source of error relates to severing of roots at the start of *in situ* field incubations, which may lead to a disturbance of rhizosphere processes, i.e. N turnover associated with root exudation and decomposition. Therefore, understanding and quantifying rhizosphere processes could lead to reduced errors in estimates of N supply in Brazilian eucalypt plantations.

There is speculation that rhisphere processes might be a signficant source of N supply in for some trees (Grayston et al., 1997), as trees create environments more favorable to microbial activity than occur in bulk soil. This effect is mainly due to the release of carbon to the soil in the form dead roots or rhizodepositions (secretions, lysates, gases, mucilages, etc). Therefore, the effect of the plant on biological activity in the rhizosphere may be important for the prediction and measurement of biological phenomena like net N mineralization in a range of ecosystems. Finzi et al. (2015) estimated that the mineralization in rhizosphere soil of temperate forests can represent 1/3 of all mineralized N in the ecosystem. This high rate of N supply from rhizosphere processes is explained by exudates released by tree roots that include carbohydrates, amino acids, organic acids, fatty acids, phenolic acids, vitamins, volatile compounds and growth factors (Grayston et al., 1996) serving as substrates for the growth of soil microbiota and their production of enzymes (Drake et al., 2013). This effect of carbon addition on microbial behaviour and, consequently, on SOM mineralization, is popularly known in the scientific literature as the priming effect, which is described in detail for soil under *Eucalyptus* by Derrien et al. (2014).

Hurtarte (2017), in a study under greenhouse conditions, observed that in the rhizosphere of *Eucalyptus* seedlings contains significant amounts of citric, malic and oxalic acids, as well as sucrose, alose, fructose, glutamine, inositol and asparagine. The author found that the release of these organic compounds was associated with decreased total N concentration in the rhizosphere, suggesting a nutritional benefit for the *Eucalyptus* seedlings. Despite these advances in rhizosphere research, there are no quantitative studies examining the importance of the priming effect in *Eucalyptus* plantations or native forests.

In relation to plant systems in general, Schimel and Weintraub (2003), Allison et al. (2010) and Drake et al. (2013), developed the MSNiP general model to estimate N rhizospheric cycling. In this model, mineralization rates depend on system stoichiometry and soil temperature. However, to improve the application of this model, it needed to be linked to plant growth and root development, as well as microbial population dynamics as affected by water, nutrients and other soil properties.



The objectives of this work were to (1) propose a model for estimating *Eucalyptus* rhizosphere cycling, (2) evaluate model performance and input sensitivity, and (3) estimate the potential importance of rhizosphere priming on N supply in a typical *Eucalyptus* plantation in Brazil.

## 2 Methods

### 2.1 ForPRAN theoretical model

The Forest plantation rhizospheric available nitrogen (ForPRAN) is a model based on the laws of conservation of matter and energy and on the principle that systems seek self-organization as a strategy of self-preservation. One of these strategies is cooperation between organisms for mutual benefit (mutualism). In this case, trees release organic compounds that modulate the rhizospheric microbial processes. The release of organic compounds into the rhizosphere provides energy and

10 labile nutrients - factors in greater abundance for it and scarce for microbiota - and receives in return a higher amount of the mineralized N and other nutrient of the soil organic matter. This symbiosis involves three components of the ecosystem - *Eucalyptus*, microbiota and soil – which may have evolved to ensure nitrogen and energy fluxes in the forest ecosystem were maintained. The process is schematically summarized in Fig. 1.

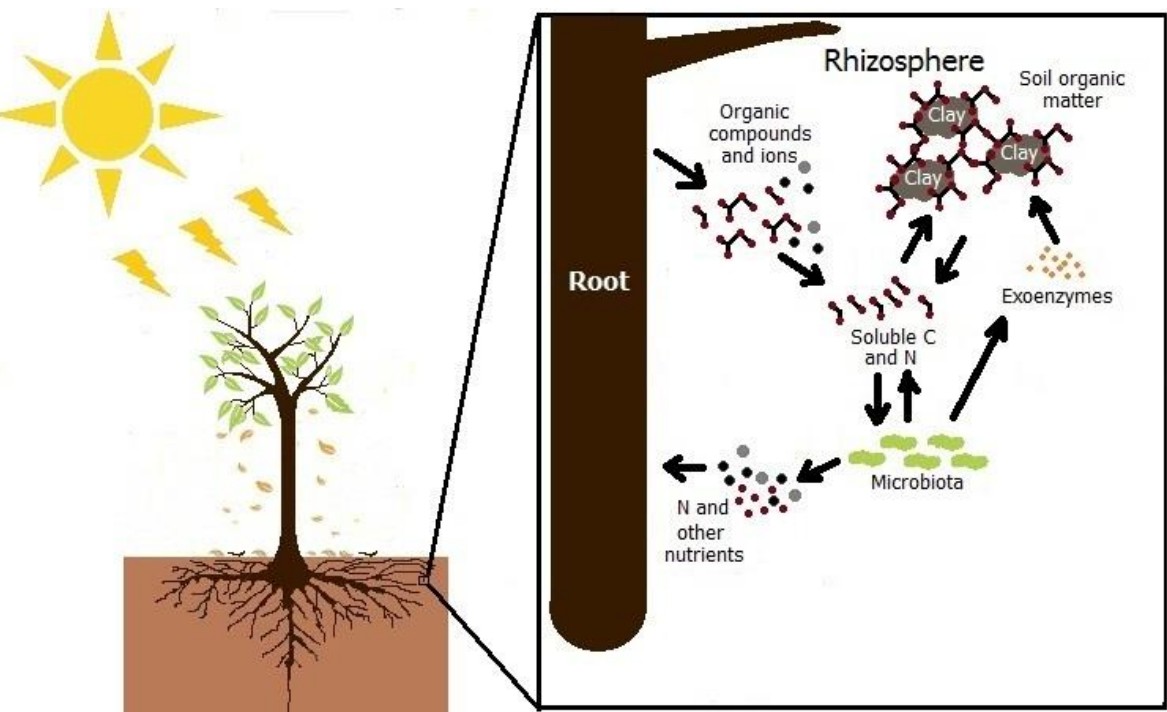

**Figure 1.** Illustration of rhizosphere C and N cycling processes in the ForPRAN model

Rhizospheric N supply is described as a function of key variables that reflect the complexity of the N cycle in the rhizosphere. These variables can be grouped into three categories, which are related to: 1- rhizosphere dimensions (root diameter; rhizosphere thickness; clay content; soil layer considered; dry matter of aerial part); 2- C and N availability and





microbial demand and metabolism (C radicular efflux rate; soil organic matter content; rhizodeposition C/N ratio; soil C/N ratio; enzymes C/N ratio; microbiota C/N ratio; soil protection capacity); 3- conditions that affect microbial turnover (total porosity, moisture and temperature).

### 2.1.1 Rhizosphere dimensions

The size of the rhizosphere soil volume is one of the most important factors influencing the priming effect. In the logic of ForPRAN, rhizosphere volume is related to the length of fine roots and their diameter and also to the thickness of the rhizosphere. The length of roots is strongly related to shoot biomass, as the source of photo-assimilates (Mello et al., 1998; Neves, 2000; Leles, 2001; Teixeira et al., 2002; Gatto et al., 2003; Maquere, 2008). Root length is also related to soil clay content, because of the relationship with availability of water and nutrients. Length is also related to the depth of the soil layer in which N priming is considered, and is usually concentrated in the top 30 cm of soil (Mello et al., 1998). The thickness of the rhizosphere depends on the nature and amount of rhizodeposited compounds (Finzi et al., 2015) as well as the soil matrix. For simplicity, ForPRAN considers a constant user-specified value of thickness according to observational data presented in Jones (1998), Barber (1995), Sauer et al. (2006) and Hurtarte (2017).

### 2.1.2 C and N availability and microbial demand

The model follows the logic of stoichiometric balance between substrate supply and microbial demand and the 'Law of the Minimum' applied to the microbial processes, as presented by Schimel and Weintraub (2003), Allison et al. (2010) and Drake et al. (2013). In general, the model considers that an increase in the availability of organic substrates increases microbial biomass and enzyme production, and therefore the processes related to soil organic matter mineralization. Microbial processes are affected in different ways according to the availability of organic C and N. For instance, when the availability of N exceeds microbial demand, it`s considered that C becomes a limiting factor, leading to an increase in net N mineralization and C immobilization. On the other hand, when C availability exceeds microbial demand, N becomes the limiting factor leading to an increase on respiration and N immobilization. Substrate availability for these processes is modulated by soil protection that in turn depends mainly on the amount of clay, mineralogy and of soil carbon content (degree of saturation of clays by organic carbon). If the soil has higher protection capacity, the microbiota is more likely to not have its nutritional demand satisfied and therefore be limited in biomass growth (Silva et al., 2011). On the other hand, if soil has minimal C and N protection, these resources are more likely to be affected by microbial attack (Silva et al., 2011).





### 2.1.3    Factors that affects microbial turnover

Soil moisture affects microbial metabolism because of its role as a universal solvent, i.e. all microbial reactions depend on water (Brock and Madigan, 1991). Therefore, there is a positive effect of moisture increase on microbial processes, being one of the most important factors in tropical environment because of the great variation in its availability throughout the year. In conditions of low water availability, microorganisms expend more energy adapting to their electrochemical condition, often by synthesizing proline and glutamine or by taking up $K^+$ (Brock and Madigan, 1991). However, it is well known that such mechanisms do not always compensate for water deficit, leading to reduce microbial biomass, as observed in the work of Sato, Tsuyuzaki and Seto (2000).

Temperature is another important factor affecting microbial metabolism, that operates in two opposing ways. Rising temperatures are responsible for elevation the rates of chemical and enzymatic reactions (Brock and Madigan, 1991). Such increases have a positive impact on microbial biomass and, therefore, are related to increases in $CO_2$ evolution and N mineralization rates (Brock and Madigan, 1991). On the other hand, above a certain temperature it`s expected that microbial cellular components are denatured (like exo-enzymes), causing microbial process rates to fall sharply (Brock and Madigan, 1991).

In ForPRAN, physical conditions affect microbial communities via porosity. Extremes of porosity reduce microbial biomass and, consequently the C and N mineralization (Silva et al., 2011). This change occurs because soil porosity affects the concentration and transport of $O_2$ (Torbert and Wood, 1992), as well as the liquid and solute movement and C and N protection by the soil matrix (Silva et al., 2011).

### 2.2  Mathematical model overview

ForPRAN is based on previously developed and evaluated functions and also on functions developed and parameterized in the present work. We used data from literature to make the model a mathematical representation of reality explained by the ForPRAN theoretical model. The model has two parts that work simultaneously: part 1 – module of fine root growth and rhizodeposition release; part 2 – module of C and N in the *Eucalyptus* rhizosphere (Fig. 2).

In the first part, we used the 3-PG model (Landsberg and Waring, 1997) to represent the conversion of light energy to dry vegetable matter. To represent the growth of fine roots, we proposed a non-linear model and adjusted it to the data presented by Mello et al. (1998), Neves (2000), Leles et al. (2001), Teixeira et al. (2002), Gatto et al. (2003) and Maquere (2008). From this point, we estimate the rate at which C and N release processes from roots as an adaptation of the model by Personeni et al. (2007) that was based on Farrar et al. (2003).

In the second part of the model, we described the rhizospheric cycling system. To do so, we modified the MSNiP model developed by Schimel and Weintraub (2003), Allison et al. (2010) and Drake et al. (2013), what consisted in considering



the effect of soil moisture, physical conditions, temperature (effect on exo-enzymes kinetics), immigration and microbial emigration on microbial population dynamics (figure 2).

The model simulates the effect of eucalypt plantation roots on C and N cycling in the rhizosphere, with particular focus on N availability and C balance. The model does not simulate N availability or C balance in bulk soil, and changes in rhizosphere C and N do not feedback to affect plant growth. For the latter, a more complex plantation production model is required than 3-PG, which does not explicitly consider N cycling. Further details of the model are presented in the Supplementary Material section.



**Figure 2.** Flow chart of processes represented in the ForPRAN model





### 2.1 Parameter estimation

Most of the parameters present in the ForPRAN were selected based on the values of observations from previous studies. For instance, parameters used for modelling fine root growth and rhizodeposition were based in several studies, which include Mello et al. (1998), Neves (2000), Leles et al. (2001), Teixeira et al. (2002), Gatto et al. (2003), Maquere (2008) and Personeni et al. (2007). The remainder parameters used for simulating C and N cycling in the rhizosphere soil (bacteria + fungi) was based mainly in the studies of Schimel and Weintraub (2003), Allison et al. (2010) and Drake et al. (2013). In addition, to the parameters used in those studies, it was used data presented in Sato et al. (2000), Neergaarda and Magid (2001), Silva et al. (2011) to estimate parameters of the models used to measure the effect of water, soil organic matter and soil physical conditions on microbial dynamics. A detailed presentation of the parameters used and their respective data sources is presented as supplementary material.

### 2.2 Evaluation of the rhizosphere model

The following are the main statistics used to describe the performance of the model in predicting microbial behavior under different treatments presented in Drake et al. (2013). The experiment of Drake et al. (2013) measured microbial biomass inclueed after daily pulse of water, water+C and water+C+N during early summer.

1- It was used a linear regression of predicted (P) microbial biomass vs observed (O) data and its coefficient of determination ($R^2$), being tested $\beta 0$ (H0=0) and $\beta 1$(H0=1).

2- Nash Sutcliffe efficiency (NSE), which describes the relative magnitude of the residual variance compared to the measured data variance (Moriasi et al., 2007).

$$NSE = 1 - \frac{\sum_{i=1}^{n}(P_i - O_i)^2}{\sum_{i=1}^{n}(O_i - \bar{O})^2}$$ 

Eq. 1

3- Mean Error (ME), which indicates any bias in the predictions.

$$ME = \frac{1}{n}\sum_{i=1}^{n}(P_i - O_i)$$ 

Eq. 2

4- Mean Absolute Error (MAE), which provides a simple description of the magnitude of estimation errors.

$$MAE = \frac{1}{n}\sum_{i=1}^{n}|P_i - O_i|$$ 

Eq. 3

5- Root Mean Square Error to Standard Deviation Ratio (RSR), which provides a standardised value of the root mean square error.

$$RSR = \frac{\sqrt{\sum_{i=1}^{n}(O_i - P_i)^2}}{\sqrt{\sum_{i=1}^{n}(O_i - \bar{O})^2}}$$ 

Eq. 4

6- A qualitative evaluation was presented considering the relationship between the increases on root exudation effect on microbial biomass vs the exo-enzymes production, respiration and total N of soil.

### 2.1. Sensitivity analysis of the ForPRAN model

In the sensitivity analysis, each variable was increased and decreased, while keeping others constant. In this way, the effect of each input variable on the response variable (e.g, N availability) was estimated. The range of variables values were selected as commonly occurring in the field. The sensitivity analysis was standardized using equation 5 (Allison et al., 2010).

$$Sensitivity\ (S) = \frac{\log(higher\ output) - \log(lower\ output)}{\log(higher\ input) - \log(lower\ input)}$$ 

Eq.5



## 3    Results and Discussion

### 3.1    Statistical parametrization and evaluation of the model

**3.1.1    Root length**

The empirical model partitioning dry matter to fine roots (<= 3 mm) had independent variables of clay content of the soil, thickness of the layer and shoot dry matter and was based on data presented in Mello et al. (1998), Neves (2000), Leles (2001), Teixeira et al. (2002), Gatto et al. (2003) and Maquere (2008). Despite the uncertainties of modeling root growth, this model had a satisfactory fit ($R^2$ = 0.75; Fig. 3). The intercept was not significantly different from 0 and the slope of 1.01 was

not significantly different from 1.

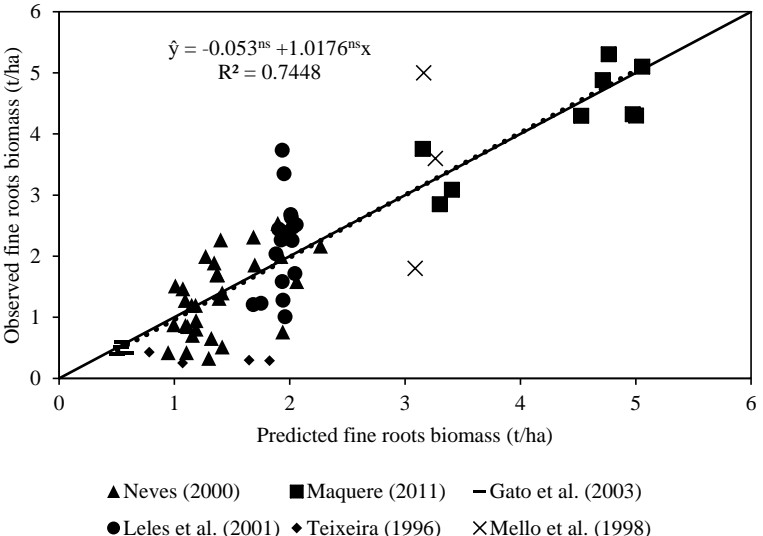

**Figure 3.** Regression of predicted versus observed fine root biomass (broke line) in relation to the 1:1 line (solid line). $\beta 0$ (H0=0) and $\beta 1$(H0=1) were evaluated by t test.

**3.1.2    Rhizosphere processes**

Figure 4 represents the main quantitative results related to the evaluation of the model. In the validation process we assumed a substrate use efficiency value of 0.3 µg µg$^{-1}$ (SUE, Table 2 of Supplementary Materials) for conditions of low availability of C and N. For higher N availability, we assumed a more efficient use of C (SUE = 0.35 µg µg$^{-1}$). We also assumed

a low value of enzymatic production rate of 0.0075 µg C µg$^{-1}$ h$^{-1}$ (Kep, Table 2 of Supplementary Materials) in the absence of C and N, while in the presence of C and N this value was considered intermediate of 0.0125 µg C µg$^{-1}$ h$^{-1}$ and in the presence of C (in the absence of N) of 0.02 µg C µg$^{-1}$ h$^{-1}$. This range was used to reflect more investment in enzymes to try to meet the microbial demand for N, when C is not the most limiting nutrient.

The model satisfactorily simulated microbial biomass across the range of observed data (Figure 4); the intercept was

not significantly different from 0 and the slope of 0.89 was not significantly different to 1, with $R^2$ = 0.91, NSE = 0.90. Mean Error (ME) (0.02) and Mean Absolute Error (MAE) (1.77) indicate that the error associated with predictions was low considering the range of the observed values (19.7-38.2 µg C/g). Root Mean Square Error to Standard Deviation Ratio (RSR)



value was of 0.32, which is low according to Moriasi et al. (2007), who suggests that values of RSR below 0.6 indicate good to very good performance. The figure 5 is a simulation of a trial performed by Drake et al. (2013) using ForPRAN model, where peaks of microbial growth are observed as a function of the addition of substrates. Microbial growth is higher in the presence of C + N, followed by only C and water.

5      Figure 6 shows that the microbial behavior predicted by ForPRAN when microorganisms receives a source of carbon. As carbon availability increases, biomass increases, which is directly proportional to increased exo-enzyme production and respiration. Conversely, when the microbial biomass increases there is a tendency of the total organic N of the soil to reduce - condition in which the decomposition of the native organic matter of the soil can surpass the formation of new SOM. Considering the high carbon investment in roots by *Eucalyptus* plants, the rhizosphere cycling model should be considered for

10    adaptation to other forestry and agricultural production models where the inclusion of such processes offer the potential for improved model performance.

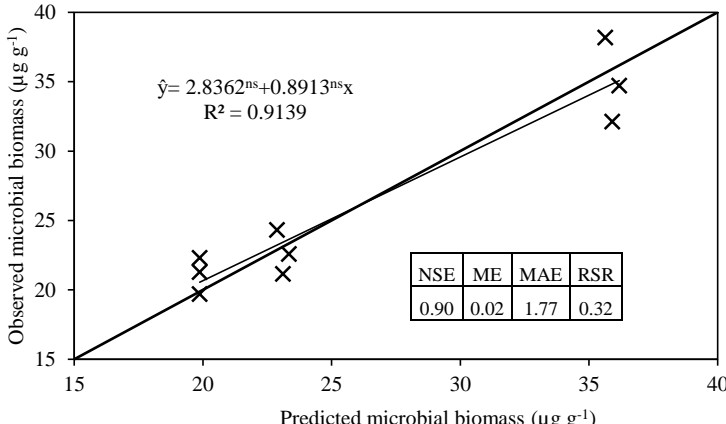

**Figure 4.** Regression of observed data (Drake et al., 2013) as a function of those predicted by the ForPRAN model. $\beta 0$ (H0=0) and $\beta 1$(H0=1) were evaluated by t test

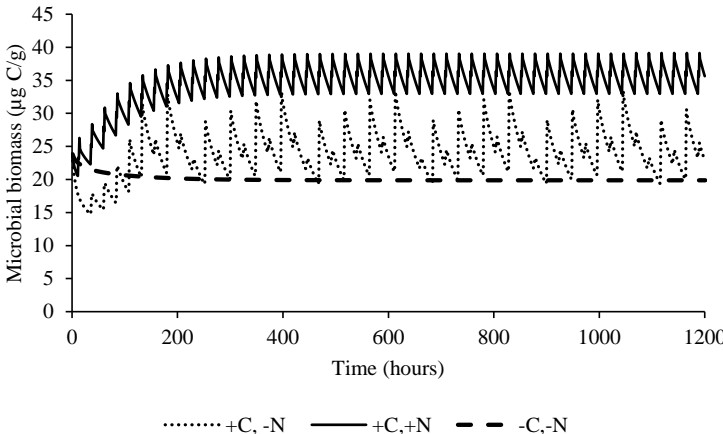

**Figure 5.** Predicted effect of daily pulses of substrates containing water, water+C or water+C+N (as occurred in Drake et al 2013) on microbial biomass during 50 days (1200 hours)



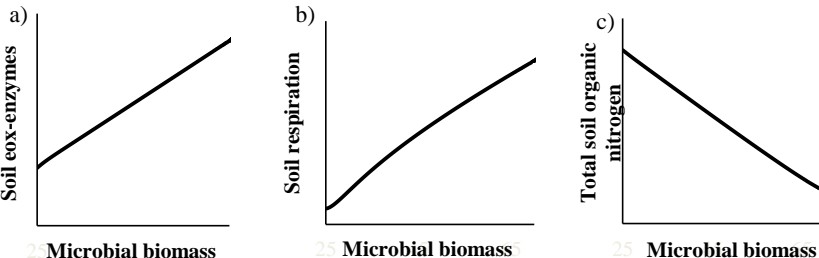

**Figure 6.** General trends of a) soil exo-enzymes b) soil respiration and c) total soil organic nitrogen as a function of microbial biomass under conditions of increasing availability of C and N

## 3.2 Model sensitivity analysis

### 3.2.1 Modeling fine root growth and rhizodeposition

The length of fine roots is of high importance for rhizospheric processes, because it partially defines the volume of the rhizosphere According to the sensitivity analysis root diameter had a major influence on root length, with lesser importance due to soil clay content, layer depth, and shoot mass (Table 1). Volume of rhizosphere had similar sensitivity, except that the thickness of the rhizosphere was the second variable that most influenced total volume, followed by root diameter (Table 4).

For a given root system, the larger the diameter considered to have a rhizosphere effect in the range 0-3 mm, the greater the estimated total root length and the larger the rhizospheric volume. On the other hand, when clay content was varied, an inverse relationship was observed with the root length response variable. For soils with lower clay content, the model estimates higher values of fine root length and, consequently, rhizosphere volumes - and vice versa (table 1). Jones (1983) the same relationship, attributing to the fact that soils with higher clay contents also presented lower values of critical density for root growth. In the case of the input variable thickness of the soil layer within a given soil profile, there is a direct relation, so that when increasing soil layer thickness also increases the length of fine roots and the volume of rhizosphere. In such a case though, three would be less soil depth (and rhizosphere volume) remaining in the rest of the profile. Finally, when shoot mass was varied, there was also a direct relation with root length and rhizosphere volume. Although these qualitative changes to rhizosphere volume in the sensitivity analysis were therefor logical, this analysis provides an indication of the relative quantitative importance of each of the inputs analysed.

It is interesting to note that root length in the average conditions (standards) was 17308 km ha$^{-1}$, for a stand with 140 t ha$^{-1}$ of aerial part dry matter mass, soil with 30 dag kg$^{-1}$ (or %) of clay, layer of depth of 0-25 cm and roots up to 1 mm in diameter (table 1). The minimum length observed in the sensitivity analysis was related to the variable entry diameter of 0.25 mm (1069 km ha$^{-1}$); the observed maximum length occurred with clay of 10 dag kg$^{-1}$ (47,555 km ha$^{-1}$), remaining unchanged the other variables. Melo et al. (1998) found values ranging about (for top 30 cm of soil) from 48,934 to 37,170, which varied greatly according to the productive characteristics of the genetic material and the type of propagation adopted. In the case of the rhizospheric volume, for the our standard conditions of a rhizosphere of 0.5 mm, the volume of the forest soils that represent this system can be equal to 135,937.5 dm³, that is, approximately 5.4 % of the forest soil. Similarly, the lowest value of rhizosphere volume was observed when roots with a diameter less than 0.25 mm (<1 % of the forest soil) were considered. The highest value observed was when the limit diameter of 3 mm was entered, with a value of 461,767 dm³ (18.5 % of the forest soil).

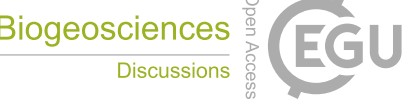



The value of the rhizospheric soil volume simulated by ForPRAN does not deviate from the estimates of Finzi et al. (2015), according to which the volume occupied by the rhizosphere of temperate forests is between 5 and 25 % of the total volume of soil. Because most of the root lengths concentrate on the surface horizons (about 70 %) (Mello et al., 1998), we consider the simulated values to be an appropriate approximation. Much of the volume of rhizospheric soil is determined by the length and architecture of the roots and the thickness of the rhizosphere (Finzi et al., 2015). As root length here was determined based on field measurement work (Mello et al., 1998; Neves, 2000; Leles, 2001; Teixeira et al., 2002; Gatto et al., 2003; Maquere, 2008), greater remaining uncertainty about the volume of rhizosphere would be related to its thickness. This, in turn, depends on the amount and nature of rhizosphere deposits, and on the physical, chemical and biological properties of the soil (Finzi et al., 2015). Default rhizospheric thickness in FORPRAN was 5 mm, being this is somewhat conservative, as literature values are: 0.2-1 mm (Jones, 1998), 2-12 mm (Sauer et al., 2006) and up to 20 mm (Barber, 1995). This aspect might be important for future in ForPRAN, whereby carbon transport models from the root surface would need to be considered that adequately accounts for soil properties.

During release of rhizodeposits, using the model of Personeni et al. (2007), it was noted that there were certain simplifications of the process, i.e. after 8 hours deposition of C and N by roots was assumed to be maximized at 7.5 and 0.75 $\mu$g cm$^{-3}$ h$^{-1}$, respectively, with no change thereafter. In nature, this value can be altered as a function of the source-sink relations in the plant, by the state of development of roots (Finzi et al., 2015), and by the physical and chemical conditions of the soil, such as the availability of P and the presence of Al (Farrar et al., 2003).

In order of decreasing importance, the variables that most influenced the total amount of C rhizodeposits were: root diameter> rhizosphere thickness = root efflux rate> clay content> soil layer thickness considered> part dry matter mass aerial part (table 2). After 10000 hours for a stand of 140 t ha$^{-1}$ of shoot (70.42 t ha$^{-1}$ of C) 1274.4 kg ha$^{-1}$ of C in rhizodeposits was estimated, i.e. 1.8 % of the primary net productivity of the shoot. The minimum value of the rhizodeposition observed in the sensitivity analysis was verified when only roots with a diameter equal to or less than 0.25 mm (19.7 kg ha$^{-1}$ of C, or 0.02 % of the net primary productivity) were considered. The highest value observed was when all roots with diameter up to 3 mm (4329 kg ha$^{-1}$ of C, or 6.15 % of the net primary productivity of the shoot) were considered, which demonstrates the influence of root length on the calculation of the rhizodeposition.

As far as a typical *Eucalyptus* plantation is concerned, the only approximation we have at an ecosystem scale is the study of Aoki et al. (2012), who studied soils with low levels of P and with range of species, among them the Myrtaceae. The estimates above are in general agreement with Aoki et al. (2012), who showed that species of Myrtaceae, represented by the genera *Syzygium* and *Tristaniopsis*, had exuded large amounts of C in the form of organic acids, i.e. c. 16.6 % of shoot mass. The above authors attributed this remarkable exudation capacity to high specific surface root area, to the high number of root apices and also to the ability of the plant to exude more in soils poor in P. For comparison purposes, the values considered normal for trees represent 40 % (van Veen et al., 1991) to 73 % of net photosynthesis invested in roots and associated microorganisms (Fogel; Hunt, 1983). The ForPRAN model estimates for longer simulations corroborate this information (data not presented).

### 3.2.2 Modeling C and N cycling in the rhizosphere soil (bacteria + fungi)

According to our model, the variables that most influenced the population dynamics of rhizospheric microbiota were the following (in order of decreasing importance): soil porosity > soil temperature > soil organic matter > radicular efflux rate (table 2). Under standard conditions, average microbial biomass estimated by the model is about 53 $\mu$g C per gram of dry soil. The maximum observed value was 142 $\mu$g/g, when soil temperature was increased from 5 to 35 °C. The main effect of temperature was on exo-enzymes kinetics, as described by the KappaD variable in figure 7, which is based on Brock and Madigan (1991). Enzymatic activity in ForPRAN is maximized between 25 and 40 ° C (Figure 7).





We use data presented Silva et al. (2011), for Oxisol soils to establish a relationship between porosity and ideal conditions for microbial growth (Figure 8). This empirical relationship led us to conclude that porosity values close to 0.53 cm³/cm³ were most favorable for the survival of microorganisms (using soil particle density of 2.60 g/cm³). The effect of soil moisture on microbial biomass was based on the work of Sato, Tsuyuzaki e Seto (2000). More pronounced limitations on microbial biomass were observed under soil moisture conditions below 40 % of field capacity (figure 9).

Increasing the supply of C and N in the rhizosphere led to growth of the rhizospheric microbial population in ForPRAN simulations. It was observed that the model simulates values with a mean of 53 μg/g (or μg/cm³ for 1 g/cm³ soil density), which corresponds to the values presented in the second quartile of 206 field observations of *Eucalyptus* plantations of southeastern Brazil (Figure 10). When temperature and the amount rhizodeposited carbon in the rhizosphere was reduced, the population decreased (table 2) and became more similar to the populations represented by the first and second quartiles of Figure 10. Silva et al. (2010) reported mean values of 358 μg microbial biomass per g soil, for the 0-20 cm depth of a Yellow Red Latosol planted with *Eucalyptus* in the region of Campos das Vertentes, MG state, brazil, which was higher than the maximum values observed in the sensitivity analysis (153 μg/g). Gama-Rodrigues et al. (2008) observed microbial biomass (carbon) of 80.6 μg/g (Aracruz / ES), 310.2 μg/g (Guanhães / MG), 95.3 μg/g ( Luís Antônio / SP), 62.4 μg/g (Lençóis Paulista / SP). Therefore, values of microbial biomass vary significantly with forest site, which highlights the importance of integrating nutrient cycling models with plant growth models and studies of the physiological behavior of different *Eucalyptus* genetic materials. Likewise, the ForPRAN model provides different estimates according to site conditions.

As a result of biological activity in the rhizosphere, average values of mineralized N (accumulated for the 10000 h period) of about 87.8 kg ha$^{-1}$ were observed (table 2). The maximum value observed was about 300 kg ha$^{-1}$, considering the sum of root rhizospheres in different classes of diameters (0 to 3 mm). The minimum value was about 1.4 kg when cycling in the rhizosphere of roots with a diameter of up to 0.25 mm was considered. The variables that most influenced this process were (in descending order of importance): root diameter > rhizosphere thickness > soil temperature > clay content (table 2). Finzi et al. (2015) estimated that N mineralization in rhizosphere soil of temperate forests can represent 1/3 of all mineralized N in the ecosystem. Our work corroborates the idea that rhizosphere processes are quantitatively important, which might also explain why in some soils supporting *Eucalyptus* plantations there is a decrease in organic matter content (Pulito et al., 2015). In complex systems, changes in factors could act individually or in combination, e.g. soil or climatic factors, and result in changes in soil organic matter. Such changes could potentially be simulated through further improvements to the ForPRAN model, e.g. by combining it with a more complex plant production and soil model such as APSIM (Holzworth et al. 2014).

The balance of inorganic N in the system variable [ΔN = (Inorganic-N Vrhizo) - (N rhizodeposited Vrhizodeposition)] expresses the nutrient gain by the plant as a result of interaction with the soil microbiota (Table 2). The actual gain of N by the plant (assimilated) is lower than the mineralized values presented previously because the plant has to release some N to induce the priming effect. The balance for standard conditions for a 10,000 h simulation was of 24.15 kg N ha$^{-1}$. The maximum value observed in the sensitivity analysis was reached at a temperature of 35°C (228.15 kg ha$^{-1}$) (table 2). The minimum value observed was when rhizodeposition occurred at a C/N ratio of 5, which led to a negative balance of about -26 kg ha$^{-1}$ for the *Eucalyptus* plant (Table 2), i.e. the plant released more N than I took up as a result of rhizosphere priming. The input variables that most influenced N balance were (in descending order of importance): soil temperature> soil C/N ratio > soil protection capacity > rhizodeposition C/N ratio.

### 3.2.3 Biological meaning

Table 3 shows a scenario simulation in which was studied the effect of two conditions of competition between microbiota and soil by carbon and nitrogen compounds on the potential supply of rhizospheric N for *Eucalyptus* cultivation over a period of seven years. It was considered for this simulation the standard conditions presented in in the sensitivity analysis





(table 1 and 2), a rhizosphere of 3 mm (Hurtarte, 2017) and all root lengths from 0 to 3 mm in diameter (total fine roots). It was observed that in soils with high carbon and nitrogen protection power there was a lower or negative potential for the rhizospheric N supply (or N gain). Under these conditions, *Eucalyptus* plants would be expected to be more responsive to nitrogen fertilization. However, under conditions of lower carbon and nitrogen protection (15 %) and 4 % SOM, rhizospheric

5    supply can contribute significantly to the N balance. For these conditions, which are speculative, the process had a positive balance for the plant: the equivalent to 24.6 % of N demand by the ecosystem (tree + litter) or 38.4 % of tree demand, which assumed losses of 40 % due to leaching, denitrification and volatilization.





**Table 1.** Values of the input variables used in the model to estimate fine root length, rhizosphere volume, C rhizodeposition

| Name | Input | | | Length (x10³ km ha⁻¹) | | | | Rhizosphere volume (x10³ dm³) | | | | C rhizodeposition (x10³ kg ha⁻¹) | | | |
|---|---|---|---|---|---|---|---|---|---|---|---|---|---|---|---|
| | Mean | Lower | Higher | Mean | Lower | Higher | S[1] | Mean | Lower | Higher | S[1] | Mean | Lower | Higher | S[1] |
| Clay content | 30 | 10 | 50 | 17.3 | 47.6 | 10.8 | 0.90 | 135.9 | 373.5 | 85.0 | 0.90 | 1.3 | 3.5 | 0.8 | 0.92 |
| Soil layer considered | 25 | 5 | 50 | 17.3 | 6.4 | 26.6 | 0.60 | 135.9 | 50.1 | 208.9 | 0.60 | 1.3 | 0.5 | 2.0 | 0.62 |
| Rhizodeposition C/N ratio | 20 | 5 | 60 | 17.3 | 17.3 | 17.3 | 0.00 | 135.9 | 135.9 | 135.9 | 0.00 | 1.3 | 1.3 | 1.3 | 0.00 |
| Root diameter | 1 | 0.25 | 3 | 17.3 | 1.1 | 19.6 | 1.20 | 135.9 | 2.1 | 461.8 | 2.20 | 1.3 | 0.02 | 4.3 | 2.17 |
| Dry matter of aerial part | 140 | 40 | 280 | 17.3 | 13.6 | 19.7 | 0.20 | 135.9 | 107.1 | 155.1 | 0.20 | 1.3 | 1.0 | 1.5 | 0.19 |
| Soil moisture | 50 | 5 | 100 | 17.3 | 17.3 | 17.3 | 0.00 | 135.9 | 135.9 | 135.9 | 0.00 | 1.3 | 0.1 | 2.6 | 0.00 |
| Enzymes C/N ratio | 5 | 3 | 7 | 17.3 | 17.3 | 17.3 | 0.00 | 135.9 | 135.9 | 135.9 | 0.00 | 1.3 | 1.3 | 1.3 | 0.00 |
| Microbiota C/N ratio | 7 | 3.5 | 14 | 17.3 | 17.3 | 17.3 | 0.00 | 135.9 | 135.9 | 135.9 | 0.00 | 1.3 | 1.3 | 1.3 | 0.00 |
| Soil C/N ratio | 12 | 6 | 30 | 17.3 | 17.3 | 17.3 | 0.00 | 135.9 | 135.9 | 135.9 | 0.00 | 1.3 | 1.3 | 1.3 | 0.00 |
| Rhizosphere thickness | 0.5 | 0.1 | 1 | 17.3 | 17.3 | 17.3 | 0.00 | 135.9 | 27.2 | 271.9 | 1.00 | 1.3 | 0.3 | 2.6 | 1.00 |
| Soil organic matter content | 40 | 12 | 80 | 17.3 | 17.3 | 17.3 | 0.00 | 135.9 | 135.9 | 135.9 | 0.00 | 1.3 | 1.3 | 1.3 | 0.00 |
| C radicular efflux rate | 1.5 | 0.25 | 4.5 | 17.3 | 17.3 | 17.3 | 0.00 | 135.9 | 135.9 | 135.9 | 0.00 | 1.3 | 0.2 | 3.8 | 1.00 |
| Total soil porosity | 0.53 | 0.45 | 0.59 | 17.3 | 17.3 | 17.3 | 0.00 | 135.9 | 135.9 | 135.9 | 0.00 | 1.3 | 1.3 | 1.3 | 0.00 |
| Soil protection | 15 | 5 | 30 | 17.3 | 17.3 | 17.3 | 0.00 | 135.9 | 135.9 | 135.9 | 0.00 | 1.3 | 1.3 | 1.3 | 0.00 |
| Microbial immigration | 0.01 | 0.001 | 0.1 | 17.3 | 17.3 | 17.3 | 0.00 | 135.9 | 135.9 | 135.9 | 0.00 | 1.3 | 1.3 | 1.3 | 0.00 |

(1) Sensitivity index



**Table 2.** Values of the input variables used in the model to estimate microbial biomass, N mineralized and N balance

| Name | Value | | | BCm (μg/g soil) | | | | N mineralized (kg ha⁻¹) | | | | N balance (kg ha⁻¹) [2] | | | |
|---|---|---|---|---|---|---|---|---|---|---|---|---|---|---|---|
| | Mean | Lower | Higher | Mean | Lower | Higher | S [1] | Mean | Lower | Higher | S [1] | Mean | Lower | Higher | S [1,3] |
| Clay content | 30 | 10 | 50 | 52.84 | 52.84 | 52.83 | 0.00 | 87.87 | 241.44 | 87.87 | 0.63 | 24.15 | 66.36 | 24.15 | 0.63 |
| Soil layer considered | 25 | 5 | 50 | 52.84 | 52.84 | 52.84 | 0.00 | 87.87 | 32.40 | 135.05 | 0.62 | 24.15 | 8.90 | 37.12 | 0.62 |
| Rhizodeposition C/N ratio | 20 | 5 | 60 | 52.84 | 52.84 | 52.84 | 0.00 | 87.87 | 229.17 | 56.48 | 0.56 | 24.15 | -25.71 | 35.23 | 1.66 |
| Root diameter | 1 | 0.25 | 3 | 52.84 | 52.84 | 52.84 | 0.00 | 87.87 | 1.36 | 298.50 | 2.17 | 24.15 | 0.37 | 82.04 | 2.17 |
| Dry matter of aerial part | 140 | 40 | 280 | 52.84 | 52.84 | 52.84 | 0.00 | 87.87 | 69.26 | 100.24 | 0.19 | 24.15 | 19.04 | 27.55 | 0.19 |
| Soil moisture | 50 | 5 | 100 | 52.84 | 24.12 | 52.97 | 0.26 | 87.87 | 69.66 | 87.94 | 0.08 | 24.15 | 5.94 | 24.21 | 0.47 |
| Enzymes C/N ratio | 5 | 3 | 7 | 52.84 | 52.84 | 52.84 | 0.00 | 87.87 | 86.62 | 88.41 | 0.02 | 24.15 | 22.90 | 24.69 | 0.09 |
| Microbiota C/N ratio | 7 | 3.5 | 14 | 52.84 | 52.84 | 52.84 | 0.00 | 87.87 | 77.04 | 93.29 | 0.14 | 24.15 | 13.32 | 29.57 | 0.58 |
| Soil C/N ratio | 12 | 6 | 30 | 52.84 | 52.84 | 52.84 | 0.00 | 87.87 | 141.36 | 55.78 | 0.58 | 24.15 | 77.63 | -7.94 | 2.70 |
| Rhizosphere thickness | 0.5 | 0.1 | 1 | 52.84 | 52.84 | 52.83 | 0.00 | 87.87 | 17.58 | 175.75 | 1.00 | 24.15 | 4.83 | 48.30 | 0.68 |
| Soil organic matter content | 40 | 12 | 80 | 52.84 | 23.58 | 84.77 | 0.67 | 87.87 | 69.17 | 100.26 | 0.20 | 24.15 | 5.45 | 36.54 | 1.00 |
| C radicular efflux rate | 1.5 | 0.25 | 4.5 | 52.84 | 19.00 | 119.66 | 0.64 | 87.87 | 36.79 | 177.65 | 0.54 | 24.15 | 26.17 | -13.51 | 1.28 |
| Total soil porosity | 0.53 | 0.45 | 0.59 | 52.84 | 47.71 | 43.17 | 0.75 | 87.87 | 85.35 | 82.92 | 0.21 | 24.15 | 21.63 | 19.19 | 0.85 |
| Soil temperature | 15.5 | 5 | 35 | 52.84 | 35.66 | 142.19 | 0.71 | 87.87 | 48.65 | 291.89 | 0.92 | 24.15 | -15.06 | 228.15 | 2.83 |
| Soil protection | 15 | 5 | 30 | 52.84 | 69.20 | 34.95 | 0.38 | 87.87 | 123.01 | 52.64 | 0.47 | 24.15 | 59.28 | -11.08 | 2.38 |
| Microbial immigration | 0.01 | 0.001 | 0.1 | 52.84 | 52.05 | 60.50 | 0.03 | 87.87 | 86.92 | 97.12 | 0.02 | 24.15 | 23.20 | 33.39 | 0.08 |

(1) Sensitivity index; (2) ΔN = (Inorganic-N Vrhizo) - (N rhizodeposited Vrhizodeposition); (3) When in the presence of negative values (N balance), we sum the module of the negative value (x) plus in the lowest output (log (x+ |x|+1)) and in the higher output (z) (z+ |x|+1),

being the equation represented in the following way: $Sensittivity\ (S) = \dfrac{\log(z+|x|+1)-\log(x+|x|+1)}{\log(higher\ input)-\log(lower\ input)}$





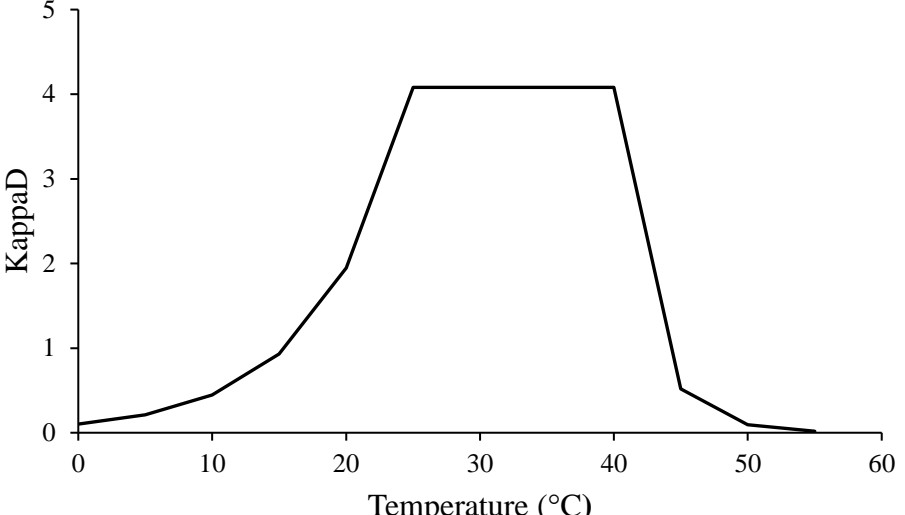

**Figure 7.** Effect of temperature on enzymatic cinetic represented by KappaD
Source: Based in Brock and Madigan (1991).

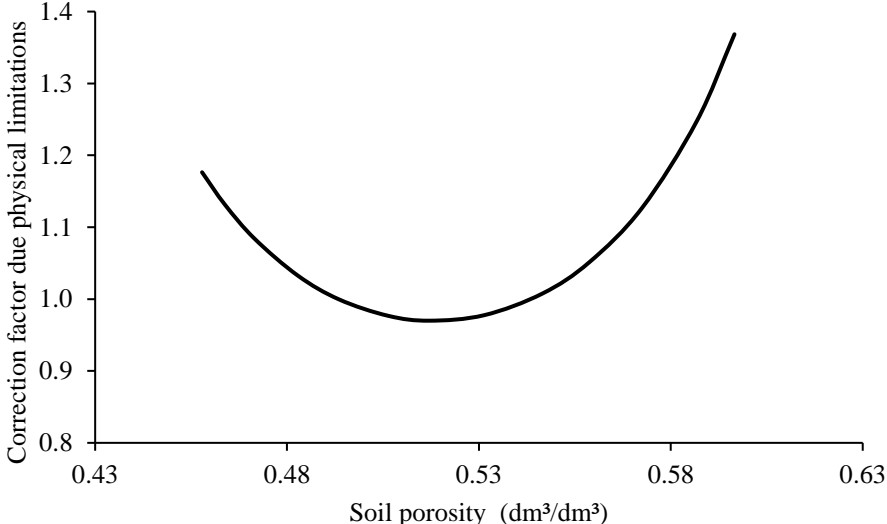

**Figure 8.** Effect of porosity on microbial death
Source: Based in Silva et al. (2011).



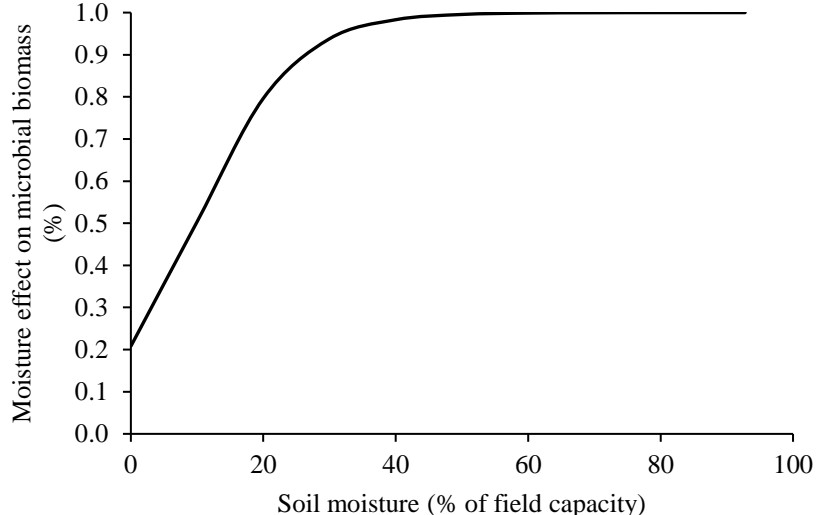

**Figure 9.** Effect of soil moisture on relative microbial biomass
Source: Based in Sato, Tsuyuzaki e Seto (2000).

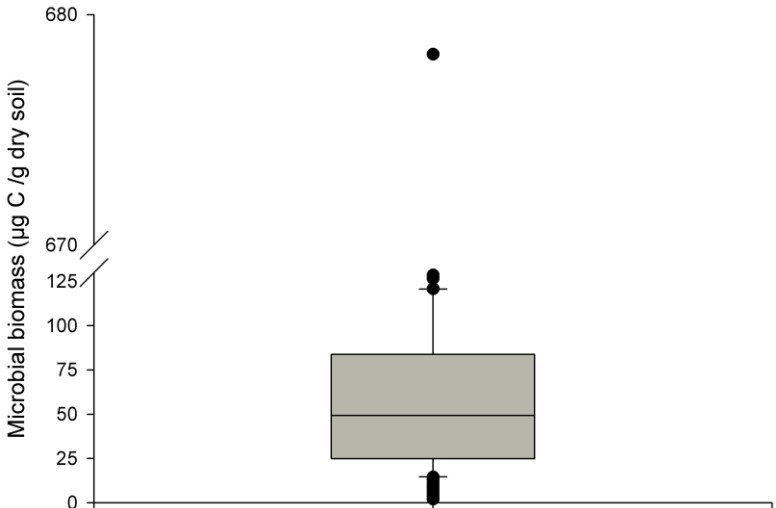

**Figure 10.** Boxplot of 206 observations of microbial biomass collected in soils under eucalyptus growing in southeast
of Brazil
Source: Data provided by Professor Marcos Rogério Tótola from Department of Agricultural Microbiology-UFV



**Table 3.** Simulation of the N gain by the priming effect in *Eucalyptus* ecosystems in two situations of soil protection of C and N

| Stand age (year) | Root length (km ha⁻¹) | Carbon and nitrogen protection by soil (%) | |
|---|---|---|---|
| | | 15 | 30 |
| 0.25 | 5191 | 13 | -7 |
| 1.39 | 10793 | 28 | -14 |
| 2.53 | 13971 | 36 | -18 |
| 3.67 | 15553 | 40 | -20 |
| 4.81 | 16502 | 42 | -21 |
| 5.95 | 16981 | 43 | -21 |
| 7.10 | 17321 | 44 | -22 |
| Cumulative rhizosphere supply | | 247 | -121 |
| Eucalyptus demand (root+shoot) | | 383 | 383 |
| Rhizosphere supply - Demand | | -136 | -504 |

## 4 Conclusion

- A model for calculating rhizospheric N priming is described that was linked a plantation production model.
- The model satisfactory preformed quantitative and qualitative in relation to observations.
- Input variables that most influenced N priming were (in descending order of importance): soil temperature> soil C/N ratio > soil protection capacity > rhizodeposition C/N ratio.
- Under the conditions presented here, the priming effect can have a significant affect *Eucalyptus* nutrition under some conditions, which can represent values of 24.6 % of the demand of the ecosystem (tree + litter) or 38.4 % of the demand of the trees, considering 40 % of losses.

## SUPPLEMENTARY MATERIAL

### Part 1 – modeling fine root growth and rhizodeposition

#### 1.1 Converting light energy in *Eucalyptus* dry matter

We used the 3-PG ecophysiological process model (Landsberg; Waring, 1997) to estimate the conversion of light energy to mass of dry matter for a mono-cultural *Eucalyptus* plantation. The role of this module is to simulate C directed to root growth and exudation in a forest plantation. To better represent the growth of plantations under tropical conditions, we used the version parameterized by Borges (2009), due to its greater degree of universality in relation to the other model parameterizations (Borges, 2012). We used shoot mass estimated by the 3-PG model as input to the next step, i.e. estimation of root length and rhizosphere volume by the ForPRAN model. For a better understanding of the equations used in the



ForPRAN model, we summarized the main variables, constants and compartments in the Table 1. The 3-PG and ForPRAN models are implemented as spreadsheets in Microsoft Excel.

**Table S1.** Variables, constants and compartments of the fine root growth and rhizodeposition model

| Name | Symbol | Unit | Default |
|---|---|---|---|
| Parameter a | a | - | 0.97 |
| Soil clay contente | Clay | dag kg$^{-1}$ (or %) | - |
| Parameter b | b | - | -0.92 |
| Parameter c | c | - | 0.62 |
| Amount of carbon released at time zero | $C_0$ | µg cm$^{-3}$ | 2.1 |
| Thickness of the soil layer considered | TSL | cm | - |
| Concentration of organic carbon in soil solution regulated by fine root | Ce | µg cm$^{-3}$ | - |
| C/N ratio of root rhizodeposition | CNrizo | - | - |
| Root length per diameter class | RLdc | cm | - |
| Specific root length | SRL | cm g$^{-1}$ | 2454 |
| Specific root length per diameter | CREd | cm g$^{-1}$ | - |
| Parameter d | d | - | 0.19 |
| Root diameter | Droot | mm | - |
| Parameter of the intercept f | f | - | 88 |
| Parameter ɣ | ɣ | - | 0 |
| Exponential decay coefficient h | h | - | 6.5 |
| Parameter of the intercept i | i | - | 20 |
| Exponential decay coefficient j | j | - | 1.6 |
| Mass of dry matter of aerial part | MDAP | t ha$^{-1}$ | - |
| Mass of dry matter of fine roots | MSfr | t ha$^{-1}$ | - |
| Mass of fine roots per diameter class | MSfrcd | t ha$^{-1}$ | - |
| Percentage of root length ratio per diameter | PAC | - | - |
| Percentage of root mass ratio per diameter | PAM | - | - |
| Root radius | r | cm | - |
| Volume of solution involving the root | V | cm$^3$ | - |
| Rate of efflux at the root apex | α | µg C cm$^{-2}$ h$^{-1}$ | 1.5 |
| Relative influx of C | β | µg C cm$^{-1}$ h$^{-1}$ | 0.2 |

Based on studies of Mello et al. (1998), Neves (2000), Leles et al. (2001), Teixeira et al. (2002), Gatto et al. (2003), Maquere (2008) and Personeni et al. (2007)



### 1.2 Estimation of carbon partitioning to fine roots (MSfr, t/ha)

An empirical model is used for partitioning of the dry matter mass to fine roots (<= 3 mm), with independent variables of clay content of the soil, thickness of the soil layer of interest, and shoot mass of the trees. The function was based on data

5   presented in Mello et al. (1998), Neves (2000), Leles (2001), Teixeira et al. (2002), Gatto et al. (2003) and Maquere (2008). We consider fine roots to be less than 2 or 3 mm, as presented by the authors. As there was no statistical difference of dry matter partition between these two diameter limits, we proposed a general model for fine roots based on 3 mm diameter.

$$MSfr = aClay^b TSL^c \mathrm{MDAP}^d \qquad \text{Eq. 1}$$

### 1.3 Estimation of the length of fine roots

To estimate the proportion of the root length in different diameters (equation 2), we assumed a sigmoidal distribution of the percentage of the total length as a function of the diameter of the fine roots, following the original proposition of Finzi et al. (2015). For example, the model for *Eucalyptus* calculated an average of 88 % of the total length of fine roots had a

15   diameter less than 1 mm (Table 1), as observed by Baldwin e Stewart (1987) and Mello et al. (1998).

$$PAC = \frac{1}{1 + fe^{-hDroot}} \qquad \text{Eq. 2}$$

### 1.4. Estimating mass partitioning to fine roots of different diameter

According to Baldwin and Stewart (1987), roots with a diameter less than or equal to 1 mm contribute more than 85% of the total length of fine roots, but there percentage of total dry matter of fine roots was much less, i.e. approximately 20% (table S1). Thus, we parameterized a sigmoidal model to represent the proportion of dry matter in relation to total root mass according to the maximum diameter considered (Droot, Equation 3). Root mass per diameter class (MSfrdc, in kg/ha) was

25   estimated by the difference in root ratio of the lower and upper diameter classes (Eequation 4).

$$PAM = \frac{1}{0,8354 + ie^{-jDroot}} \qquad \text{Eq. 3}$$

$$MSfrdc = MSfr\,(PAMn - PAMi) \qquad \text{Eq. 4}$$

30   ### 1.5 Root growth per diameter class

We used total root length in Mello et al. (1998) and equations 4 and 5 to calculate specific root length (SRL, cm/g) for a root diameter class of interest (SRLcd, cm/g) (equation 5). Root length per diameter class (RLdc, km/ha) was estimated by multiplying the root mass of the diameter class by the specific root length of the class (Equation 6).



$$SRLdn/i = SRL\frac{(PACn/i)}{(PAMn/i)} \qquad \text{Eq. 5}$$

$$RLdc = (MSrfcd\ SRLdn) - (MSrfcd\ SRLdi) \qquad \text{Eq. 6}$$

**1.6 Estimation of the rizodeposition process**

We used equation 7 to describe net rhizodeposition of carbon by the root, using a model proposed by Farrar et al. (2003) and optimized and parameterized by Personeni et al. (2007). The estimation of rhizodeposition of organic nitrogen was carried out by dividing the carbon value by the C/N ratio of the rhizodeposited material (Ne, µg cm$^{-3}$) (equation 8).

$$Ce = \frac{\alpha}{\beta(1-\gamma)\mathrm{RLdc}}\left[(\mathrm{RLdc}+1)^{1-\gamma}-1\right]\left(1 - e^{-\frac{\beta 2\pi r\mathrm{RLdc}}{V}t}\right) + \frac{C_0}{V}e^{-\frac{\beta 2\pi r\mathrm{RLdc}}{V}t} \qquad \text{Eq. 7}$$

$$Ne = \frac{Ce}{CN_{rizo}} \qquad \text{Eq. 8}$$

**Part 2 – Modeling C and N cycling in the rhizosphere soil (bacteria + fungi)**

To estimate N rhizospheric cycling, we used the model of fine root growth and rhizospheric carbon flux described above coupled to the equations of Schimel and Weintraub (2003) and Allison et al. (2010), and modified and parameterized by Drake et al. (2013) in the MSNiP model. In this model, the mineralization rates depend on stoichiometry and soil temperature. To improve the temporal and spatial resolution, we considered the plant component, as previously mentioned in the module 1, and also the population dynamics module as affected by water, nutrients and soil properties. In a very simplified way, we attribute constants to the effect of soil on the protection of the released compounds in solution, and also to the processes of microbial immigration and emigration. Table S2 lists the variables, parameters, units, and reference values used in this part of the model.

**Table S2.** Variables, constants and compartments of the microbial rhizosphere model

| Name | Symbol | Unit | Default |
| --- | --- | --- | --- |
| C in microbial biomass | BCm | µg g$^{-1}$ | 23. 817[@] |
| N in microbial biomass | BNm | µg g$^{-1}$ | 3.402[@] |
| Soil moisture | CAD | % | - |
| Enzyme C/N ratio | CNenz | µg µg$^{-1}$ | 3[@] |
| Microbiota C/N ratio | CNm | µg µg$^{-1}$ | 7[@] |
| Soil C/N ratio | CNs | µg µg$^{-1}$ | 12[@] |



| | | | |
|---|---|---|---|
| C of dead microorganisms that return to DOC | CYc | µg g$^{-1}$ | - |
| N of the dead microbiota that return to DON | CYn | µg g$^{-1}$ | - |
| Depolymerization rate of soil organic C | Dc | µg g$^{-1}$ h$^{-1}$ | - |
| Depolymerization rate of soil organic N | Dn | µg g$^{-1}$ h$^{-1}$ | - |
| Organic C in solution | DOC | µg g$^{-1}$ | - |
| Organic N in solution | DON | µg g$^{-1}$ | - |
| Density of particules | Dp | kg dm$^{-3}$ | - |
| Density of the soil | Ds | kg dm$^{-3}$ | - |
| Activation energy for absorption of DOC | Eauptake | kJ mol$^{-1}$ grau C$^{-1}$ | 47 |
| Rate of enzymatic degradation of C | ELc | µg g$^{-1}$ | - |
| Rate of enzymatic degradation of N | ELn | µg g$^{-1}$ | |
| Rate of enzyme production of C | EPc | µg g$^{-1}$ | - |
| Rate of enzyme production of N | EPn | µg g$^{-1}$ | - |
| Universal gas constant | Gasconstant | kJ mol$^{-1}$ K$^{-1}$ | 0.008314 |
| C of the dead microbiota returning to SOC | Hc | µg g$^{-1}$ | - |
| N of the dead microbiota returning to SON | Hn | µg g$^{-1}$ | - |
| Microbial immobilization rate | Jn | µg g$^{-1}$ | - |
| Decomposition constant of SOC | kappaD | - | 1 |
| Rate of enzymatic production per unit of biomass | Kep | µg µg$^{-1}$ h$^{-1}$ | 0.0005 |
| Half-saturation Michaelis-Menten constant | Kes | - | 0.3 |
| Rate of microbiota maintenance | Km | - | 0.01 |
| Temperature-dependent Michaelis constant | Kmuptake | µg C g$^{-1}$ | - |
| km of DOC uptake at 0 °C | Kmuptake0 | µg C g$^{-1}$ | 0.154 |
| Rate of increase of km uptake with temperature | Kmuptakeslope | µg C g$^{-1}$ °C$^{-1}$ | 0.015 |
| Rate of microbial biomass dying and being recycled to DOC compartments | Kr | - | 0.085 |
| Basic proportion of microbiota death | Kb | - | 0.012 |

**To be continued...**

**Table S2.** Variables, constants and compartments of the microbial rhizosphere model

| Name | Symbol | Unit | Default |
|---|---|---|---|
| Proportion of biomass dying due to water deficiency | KU | - | - |



| | | | |
|---|---|---|---|
| Proportion of DOC and DON that is protected by soil | Kpr | - | 0.15 |
| Rate of death by limitation by level of fertility | Kmf | - | - |
| Death by limitation for physical reasons | Kpt | - | - |
| Root length | L | cm | - |
| Microbial rate of mineralization | Mn | $\mu g\ g^{-1}$ | - |
| Microbial respiration rate for enzymatic production | Re | $\mu g\ g^{-1}$ | - |
| Rate of microbial respiration for growth | Rg | $\mu g\ g^{-1}$ | - |
| Maintenance respiration rate | Rm | $\mu g\ g^{-1}$ | - |
| Overflow respiration rate | Ro | $\mu g\ g^{-1}$ | - |
| Efficiency of substrate use | SUE | - | 0.3 |
| Soil temperature | Ts | °C | - |
| Rate of C uptake by microbiota | Uc | $\mu g\ g^{-1}$ | - |
| Rate of N uptake by microbiota | Un | $\mu g\ g^{-1}$ | - |
| Maximum inflow of C and N by microbiota | Vmaxuptake | $\mu g\ C\ \mu g^{-1}\ h^{-1}$ | - |
| Pre-exponential rate of C uptake | Vmaxuptake0 | $\mu g\ C\ g^{-1}\ h^{-1}$ | $1.5\ 10^{8}$ |
| Rhizosphere volume | Vrhizo | cm³ | - |
| Root mean radius | r | cm | - |
| Rhizosphere thickness | z | cm | - |
| Parameter z1 | z1 | - | 1 |
| Parameter z2 | z2 | - | 3.805 |
| Parameter z3 | z3 | - | 0.135 |

Based on studies of Schimel e Weintraub (2003), Allison et al. (2010), Drake et al. (2013), Sato et al. (2000), Neergaarda and Magid (2001) and Silva et al. (2011); [@] suggested initial values

## 2. 1 Soil organic matter (SOM) depolymerization by microbiota

The rate of depolymerization of carbon (SOC) and soil organic nitrogen (SON) to produce carbon (DOC) and nitrogen (DON) forms in soil solution is described as a Michaelis-Menten kinetic model, related to the concentration of enzymes in soil (EC) (equation 9) (Schimel; Weintraub, 2003; Drake et al., 2013). According to these authors, the depolymerization fluxes of SOC and SON (Dc and Dn) are linked by the C/N ratio of the soil (equation 9). Depolymerization would theoretically be limited by the stocks of SOC and SON, but we assumed on average that roots do not have sufficient longevity to exhaust the entire stocks of SOC and SON. Nevertheless, we consider that once the entire stock of organic matter in the soil is depleted, the microorganisms will be supplied solely by the rhizodepositions.

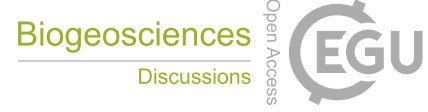

$$Dc = kappa_D \frac{EC}{Kes+EC} \qquad \text{Eq. 9}$$

$$Dn = \frac{Dc}{CN_S} \qquad \text{Eq. 10}$$

We assumed that temperature influences enzymatic kinetics by being optimal in the range 25°C to 40°C and decreasing rapidly at higher and lower values, which is consistent Brock and Madigan (1991) and Drake et al. (2013).

$$\begin{cases} if\ T \leq 25\ °C, kappaD\ =\ 0.1014e^{0.1478T} \\ \quad if\ 25 < T \leq 40\ °C, kappaD = 4.0809 \\ if\ T > 40\ °C, kappaD\ =\ 2*10^6 e^{-0.337T} \end{cases} \qquad \text{Eq. 11}$$

## 2.2 Flow of carbon and nitrogen uptake from the soil by the microbiota

The uptake of DOC and DON by the microbiota presented in Drake et al. (2013) followed the original proposal of Allison et al. (2010). The maximum velocity (Vmax) and the half-saturation constant of uptake (Km) was calculated as a function of soil temperature, according to equations 12 and 13. To estimate the soil temperature (to the depth of up to 20 cm) from air temperature, we used the daily time-step model proposed by Paul et al. (2004) for ecosystems with trees. The uptake of DOC (Uc) and DON (Un) is estimated according to the Michaelis-Menten model presented in equations 14 e 15. Uptake rates are limited by substrate availability, which means that Uc and Un cannot exceed DOC and DON, respectively (equations 16 and 17).

$$Vmaxuptake = Vmaxuptake0\ e^{-1(Eauptake \div Gasconst \cdot (T\ +273.15))} \qquad \text{Eq. 12}$$

$$Kmuptake\ =\ kmuptakeslope\ T\ +\ Kmuptake0 \qquad \text{Eq. 13}$$

$$Uc = \frac{Vmaxuptake\ BCm\ DOC}{Kmuptake+DOC} \qquad \text{Eq. 14}$$

$$Un = \frac{Vmaxuptake\ BNm\ DON}{Kmuptake+DON} \qquad \text{Eq. 15}$$

$$Uc\ = \begin{cases} Uc, & se\ Uc < DOC \\ DOC, & se\ Uc > DOC \end{cases} \qquad \text{Eq. 16}$$

$$Un\ = \begin{cases} Un, & se\ Un < DON \\ DON, & se\ Un > DON \end{cases} \qquad \text{Eq. 17}$$

## 2.3 Microbial metabolism




In the model, microbial demand considers the fact that microorganisms use C and N to synthesize exoenzymes and for the maintenance of the biomass via respiration (Schimel; Weintraub, 2003; Allison et al. 2010; Drake et al., 2013). The calculation of demand aims to determine which of the two nutrients is more limiting to the growth of the microbiota, according to equation 18. Therefore, in each step of the model, if DOC uptake does not reach a value that meets microbial demand ($Uc$), microorganisms are considered limited by C (Schimel; Weintraub, 2003; Allison et al. 2010; Drake et al., 2013). Otherwise, when $Uc$ exceeds or equals microbial demand for C, microorganisms are assumed to be limited by N (Schimel; Weintraub, 2003; Drake et al., 2013).

$$\begin{cases} Uc < Rm + \frac{EP_c}{SUE} + (Un - EP_n)\frac{CN_m}{SUE}, \text{therefore, it is limited by C} \\ Uc \geq Rm + \frac{EP_c}{SUE} + (Un - EP_n)\frac{CN_m}{SUE}, \text{therefore, it is limited by N} \end{cases} \qquad \text{Eq. 18}$$

## 2.4 Mineralization and immobilization

The immobilization rate of N ($Jn$) is zero with C limitation, or immobilization occurs under N limitation (equation 19) (Schimel; Weintraub, 2003; Allison et al. 2010; Drake et al., 2013). Microorganisms mineralize N during C limitation, but N mineralization is zero when limited by N (equation 20) (Schimel; Weintraub, 2003; Allison et al. 2010; Drake et al., 2013).

$$\begin{cases} Jn = 0, & \text{if is limited by C} \\ Jn = \left(Uc - Rm - \frac{EPc}{SUE}\right)\left(\frac{SUE}{CN_m}\right) - EPn - Un, & \text{if is limited by N} \end{cases} \qquad \text{Eq. 19}$$

$$\begin{cases} Mn = Un - EPn - \left(Uc - Rm - \frac{EPc}{SUE}\right)\left(\frac{SUE}{CN_m}\right), & \text{if is limited by C} \\ Mn = 0, & \text{if is limited by N} \end{cases} \qquad \text{Eq. 20}$$

## 2.5 Production and degradation of enzymes

It is assumed that the rate of enzyme production by the microbiota is directly proportional to microbial biomass (equation 21) and that the degradation of the enzymes is described by a constant that is multiplied by the amount of enzymes in rhizospheric soil (equation 22), as presented Allison et al. (2010) and Drake et al. (2013). Similarly, N transferred during enzymatic ($EPn$) and degradation ($ELn$) production is represented by equations 23 and 24, respectively.

.

$$EPc = Kep\ BCm \qquad \text{Eq. 21}$$

$$ELc = K_1 EC \qquad \text{Eq. 22}$$





$$EPn = \frac{EPc}{CN_{enz}} \qquad\qquad \text{Eq. 23}$$

$$ELn = \frac{ELc}{CN_{enz}} \qquad\qquad \text{Eq. 24}$$

## 2.6 Respiration process

Microorganisms use C in the respiratory process to support the maintenance of biomass (Rm) (equation 25), enzyme production (Re) (equation 26), growth (Rg) (equation 27) and "overflow" metabolism (equation 28) (Schimel; Weintraub, 2003; Allison et al. 2010; Drake et al., 2013). At this point in particular, the 'Law of the Minimum' in the respiratory process for growth is applied, so whether C or N is missing determines the magnitude of respiration.

$$Rm = Km\ BCm \qquad\qquad \text{Eq. 25}$$

$$Re = \frac{EPc\ (1-SUE)}{SUE} \qquad\qquad \text{Eq. 26}$$

$$\begin{cases} Rg = \left(Uc - \frac{EPc}{SUE} - Rm\right)(1 - SUE), & \text{if limited by C} \\ Rg = (Un - J_n - EP_n)CN_m \frac{(1-SUE)}{SUE}, & \text{if limited by N} \end{cases} \qquad \text{Eq. 27}$$

$$\begin{cases} Ro = 0, & \text{if limited by C} \\ Ro = \left(Uc - Rm - \frac{EP_c}{SUE}\right) - (Un + Jn - EPn)\frac{CN_m}{SUE}, & \text{if limited by N} \end{cases} \qquad \text{Eq. 28}$$

## 2.7 Population dynamics

In addition to the MSNiP model, the processes of microbial immigration and emigration, are represented by constant
inputs to and outputs from the rhizosphere. As for Schimel e Weintraub (2003), Allison et al. (2010) and Drake et al. (2013), there is an assumed rate (kb) of death of microorganisms each hour. However, differently from the above authors, we consider this rate for standard conditions for the survival of the rhizospheric microorganisms to be increased by a multiplicative factor (KU) under inadequate water conditions. For this purpose, we used a logistic model based on data presented in Sato et al. (2000). We also consider important that soil physical conditions affected the death of the microbiota by changes in the
availability of $O_2$, water retention and access to substrates. Hence, , we adjusted an empirical model that corrects the rate of death of microbial biomass as a function of changes in total soil porosity, according to data presented in Silva et al. (2011), where particles density of 2.6 g cm$^{-3}$ was assumed.

We also considered the effect of fertility on microbial death, based on data presented about of the difference in microbial biomass between fertile and infertile soils (Neergaarda; Magid, 2001). These modifications were the main
improvements made in the MSNiP model.



Immigration and emigration

$Im = Ki$                                                                Eq. 29

$Em = Ke$                                                           Eq. 30

Death by water limitation

$KU = (\frac{z1}{z1+z2e^{(-z3CAD)}})^{-1}$                              Eq. 31

Death by physical conditions limitations

$Kpt = \frac{1}{-12.206+ 51.060Pt -49.239\ Pt^2}$                     Eq. 32

$Pt = 1 - \frac{Ds}{Dp}$

Death by soil fertility limitations

$K_{tf} = \frac{Kb}{level\ n}$                                         Eq. 33

Level 1 (low fertility) = 1  (SOM ≤ 1.2 dag kg$^{-1}$)

Level 5 (medium fertility) = 3 (1.2 dag kg$^{-1}$ < SOM ≤ 4 dag kg$^{-1}$)

Level 10 (high fertility) = 10 (4 dag kg$^{-1}$ < SOM ≤ 8 dag kg$^{-1}$)

Final rate of microbial death

$Kmf = K_b KU_{pt} K_{tf}$                                            Eq. 34

**2.8 Internal cycling of the dead microbiota**

        The ratio (Kmf) of the C and N contained in microbiota that due death process returns to the DOC (CY$_c$) and DON
(CY$_n$) compartments is described in equations 35 and 36.

$CY_c = KmfBC$                                                        Eq. 35

$CY_n = \frac{CY_c}{CN}$                                          Eq. 36

**2.9 Module of changes in the compartments of rhizospheric C and N**

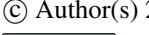



This module integrates C and N cycling in relation to rhizospheric microbiota and soil, constituting the main outputs of the ForPRAN. Changes in the different compartments are simulated over time at an hourly time-step, using equations 37-46. Another modification in relation to the MSNiP was to consider that only one proportion (1-Kpr) of the DOC and DON compartment as able to be absorbed by microbes, so that a value (Kpr DOC and Kpr DON) is protected by soil from microbial attack returning to the compartment C and N of the soil (SOC and SON).

**Table S3.** Equations used to calculate compartment changes

| N° | Compartment | Equation |
|---|---|---|
| 37 | Microbial biomass (carbon, µg/cm³) | $BCm\,(i+1) = BCm\,(i) + Uc - CYc - EPc - Ro - Re - Rm - Rg + Imc - Emc$ |
| 38 | Microbial biomass (nitrogen, µg/cm³) | $BNm\,(i+1) = BN\,(i) + Un - CYn - EPn - Mn + Jn + Imn - Emn$ |
| 39 | Enzymes (carbon, µg/cm³) | $EC(i+1) = EC(i) + EPc - ELc$ |
| 40 | Enzymes (nitrogen, µg/cm³) | $EN\,(i+1) = EN(i) + EPn - ELn$ |
| 41 | Carbon in solution (DOC, µg/cm³) | $DOC\,(i+1) = (1 - Kpr)(DOC(i) + Ce + Dc + CYc + ELc) - Uc$ |
| 42 | Nitrogen in solution (DON, µg/cm³) | $DON\,(i+1) = (1 - Kpr)(DON(i) + Ne + Dn + CYn + ELn) - Un$ |
| 43 | Soil organic carbon (SOC, µg/cm³) | $SOC\,(i+1) = SOC\,(i) - Dc + Kpr\,(DOC(i) + Ce + CYc + ELc)$ |
| 44 | Soil organic nitrogen (SON, µg/cm³) | $SON\,(i+1) = SON(i) - Dn + Kpr\,(DON(i) + Ne + CYn + ELn)$ |
| 45 | Inorganic nitrogen (µg/cm³) | $N\,(i+1) = (1\text{-loss})[N\,(i) + Mn - Jn]$ |
| 46 | N balance (kg/ha) | $\Delta N = (N\ \text{inorgânico Vrhizo}) - (N\ \text{rizodepositado Vrhizodeposition})$ |

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
