# Peer review of "Modeling rhizosphere carbon and nitrogen cycling in *Eucalyptus* plantation soil"

_Biogeosciences, 2017_

## Referee Comment (RC1) · Anonymous Referee #2 · 8 Jan 2018

General comments The study by Valadares and co-authors proposes a mechanistic model (ForPRAN) for the prediction of rhizosphere C and N cycles in Eucalyptus plantations. It is based on different belowground processes and uses (i) the 3-PG ecophysiological process model to simulate fine growth and rhizodeposition release on the basis of PAR radiation. Rhizosphere dimensions were estimated on the basis of root length and diameter. As a second part of the ForPRAN model, (ii) the stoichiometrically-constrained microbial decomposition simulation model (MCNiP model) was used to explore the dynamic relationships between organic C and microbial processes and estimate microbial N decomposition. The MCNiP model was refined by considering climatic factors as well as microbial movement. The development of modelling approaches for the prediction of rhizosphere processes and especially of rhizodeposition

and microbial priming effects is an urgent task. However, I am not completely convinced that the current model is actually approaching this task. For one, rhizodeposition is modelled from PAR, which is used for representing root length growth (P. 5, l. 27): what is the basis of the assumption that there is a direct connection between these two players? Exuded C of trees can also originate from C storage in trees. In addition, the rate of exudation is highly responsive to different environmental conditions. Secondly, priming effects seem to be a central focus of this study, but the definition of it is rather vague. Priming effects are defined as change in native C mineralization in response to increased labile C input, e.g. from rhizodeposition. Priming effects in its narrower sense describe the increase in soil respiration or N mineralization that originates neither from background respiration or mineralization nor from respiring or mineralizing the additional organic C, but rather from microbial mining of additional recalcitrant SOM. The model in its current formulation does not describe this effect. Finally, I think the general reasoning for the set-up of the study could be improved: Why is it important to derive predictions on rhizosphere C and N cycling for Eucalyptus plantations specifically? Why is rhizosphere priming important? Some language editing is also needed throughout (some typos need to be erased).

Specific comments P. 1, l. 11: The unit for fine root length is unusual, please change to m m-2 or similar. P. 1, l. 14: . . .for the prediction of rhizosphere C and N cycling. . . P. 1, l. 19: Do you mean root or soil respiration? N mineralization? N immobilization? . . .immigration, and SOM formation. P. 1, l. 21: . . .variables that influenced N gain from rhizosphere N mineralization most were. . . P. 2, l. 6: I think the expectation that additional N is expected in deeper soil layers than in the topsoil is quite surprising: Where should this N come from? N enters the ecosystem from N fixation and N cycling with leaf litter, which are both processes that are mostly occurring in the topsoil. P. 2, l. 7: . . .higher than. . . . . .observation that growth responses. . . P. 2, l. 14: . . .rhizosphere processes. . . . . .N supply for some trees. . . P. 2, l. 16: . . .in the form of dead roots. . . P. 2, l. 32: The abbreviation for the Microbial Carbon and Nitrogen Physiology model is 'MCNiP'. P. 2, l. 33-34: It remains unclear why mineralization rates need to be linked

to plant growth and root development, as well as to microbial population dynamics. This is the reasoning for the development of ForPRAN and, thus, should be given some more room. P. 3, l. 1-3: Why is it a step forward to develop a specific model for Eucalyptus from the more general (stoichiometrically-constrained) MCNiP model? P. 4, l. 5: …influencing the ecosystem priming effect… P. 5, l. 11: Responsible for elevation? Figure 2: Parts of this figure are not legible. Figure 7: kinetic Fig. 10: Eucalytpus

---

## Author Comment (AC2) · 24 Jan 2018

Dear Editor and Reviewer,

First of all, we thank the reviewer and editorial staff for comments on our manuscript. Within the possibilities, we will clarify doubts and, as far as possible, respond to the reviewer's comments. In places, it seemed appropriate to provide several paragraphs of detail and provide references.

Sincerely yours,

Rafael V. Valadares

Soil Department

[Figure]

Universidade Federal de Viçosa

36570-900 Viçosa, MG, Brazil

E-mail: rafaelvvaladares@hotmail.com

Tel: +55 (31) 99253 7720

Response to comments by Reviewer

1. Comment 1.: General comments The study by Valadares and co-authors proposes a mechanistic model (ForPRAN) for the prediction of rhizosphere C and N cycles in Eucalyptus plantations. It is based on different belowground processes and uses (i) the 3-PG ecophysiological process model to simulate fine growth and rhizodeposition release on the basis of PAR radiation. Rhizosphere dimensions were estimated on the basis of root length and diameter.

Note 1: Fine roots biomass was estimated using an empiric model developed by us and mentioned on page 21. We did not directly use PAR, but instead shoot biomass, and also soil depth and clay content. The reasons for choosing these variables is explained on the pages 4 and 11.

2. Comment 2.: The MCNiP model was refined by considering climatic factors as well as microbial movement.

Note 2: We also considered a soil protection effect on substrate availability (page 4), as well as soil porosity (page 5) and other nutritional limitations using the soil fertility index (page 28).

3. Comment 3: The development of modelling approaches for the prediction of rhizosphere processes and especially of rhizodeposition and microbial priming effects is an urgent task. However, I am not completely convinced that the current model is actually approaching this task. For one, rhizodeposition is modelled from PAR, which is used for representing root length growth (P. 5, l. 27): what is the basis of the assumption that

there is a direct connection between these two players? Exuded C of trees can also originate from C storage in trees. In addition, the rate of exudation is highly responsive to different environmental conditions. Answer 3: As mentioned in Note 1 above, fine roots biomass was estimated using an empiric model developed by us and mentioned on page 21. We did not directly use PAR, but instead shoot biomass, and also soil depth and clay content. The reasons for choosing these variables is explained on the pages 4 and 11. The links between light, photosynthesis, respiration, shoot biomass, root biomass, and exudates is well established in the literature, e.g. KUZYAKOV (2001), which explains the inner control by plants of the rhizodeposition process. Please, also see the their Figure 3, which corroborates the light-shoot-root C relationship.

When we consider Equations 7 and 8 in the Supplementary section (page 22), root length is just one element; others are the rate of efflux at the root apex ($\alpha$) and the relative influx of C ($\beta$), etc. Hence, the efflux rate can be parameterized to consider environmental factors (external control factors), like microorganisms, soil texture, nutrient availability, etc. It is important to keep in mind that proposing a model does not imply that it will be fully parameterized for all the existing variability in nature. An experimental effort will have to be made to quantify rhizodeposition under contrasting environmental conditions for the eucalypts plantations. However, the current version allows calculation of an average value for trees (NEWMAN, 1985; PAUSCH; KUZYAKOV, 2017), which is a good start. In future, with additional research and understanding, we expect improvements to the model will include additional or refined controls on rhizodeposition, but it is reasonable not expect all such details to be included in the first version of this model.

4. Coment 4: Secondly, priming effects seem to be a central focus of this study, but the definition of it is rather vague. Priming effects are defined as change in native C mineralization in response to increased labile C input, e.g. from rhizodeposition. Priming effects in its narrower sense describe the increase in soil respiration or N mineralization that originates neither from background respiration or mineralization nor

from respiring or mineralizing the additional organic C, but rather from microbial mining of additional recalcitrant SOM. The model in its current formulation does not describe this effect.

Answer 4: The priming effect (PE) was first reported by Löhnis (1926) by studying the decomposition of green manure of legume plants in soil, where there was intensified mineralization of the humus by the addition of fresh organic residues to the soil. Nowadays, it is well known that PE is common in most plant species and may be caused by other compounds besides plant residues, like dead microorganisms, high-molecular and low-molecular organic substances, or even mineral N (KUZYAKOV, 2002; BLAGO-DATSKAYA; KUZYAKOV, 2008; CHENG et al., 2014; DIJKSTRA et al., 2013; ZHU et al., 2014). Therefore, the priming effect definition that we some other authors use is: a short-term change in the SOM turnover caused by the addition of plant residues, dead microorganisms, high and low-molecular organic substances or mineral N (KUZYAKOV, Y.; FRIEDEL; STAHR, 2000). The mechanism of priming effect can be seen in the following way: 1- after the addition of a source of C/energy, the microbial succession begins, where the activation of several previously inactive microorganisms occurs and many of which respond to specific substrates (BLAGODATSKAYA; KUZYAKOV, 2008); 2- the increased activity of this microorganism increases soil organic matter degradation as a result of co-metabolism and increased enzyme production, ie, "real" priming effect (RPE) (Cadded <Cprimed >CBM) (BLAGODATSKAYA; KUZYAKOV, 2008).

Figure 1. Illustration of rhizosphere C and N cycling processes in the ForPRAN model (attached)

The figure above that illustrates our model also describes the phenomenon of co-metabolism, which is one of the most important hypotheses of the priming effect. But it is still important to know other definitions before concluding our answer: The term rhizosphere was first used by Hiltner (1904) (so, before Löhnis (1926)) to refer to the soil zone under the influence of legume roots. Current definitions of rhizosphere are generally close to this idea, with only minor modifications such as i) zone of soil immediately

adjacent to the roots of the plants, where the quantities or activities of microorganisms differ from the rest of the soil, ii) soil volume influenced by root activity, and iii) soil adjacent to the roots with a different physical, chemical and biological environment from the rest of the soil (FAGERIA; STONE, 2006). Thus, the rhizosphere is comprised of the microbiome influenced by the roots, a region that is constantly altered by organic compounds and ions and is therefore associated with a microbial population of unique characteristics of diversity and metabolism. However, the priming effect description came later than the definition of the rhizosphere effect. Thus, the description of the C and N cycle in the rhizosphere soil is enough to know that the environment is under the constant influence of C/energy and with a SOM turnover rate different to bulk soil. Considering the precedence of the description of the rhizospheric effect compared with rhizosphere priming effect on microbial dynamics and soil organic matter turnover, we can say that the rhizosphere effect definition already includes the definition of priming effect with respect to C and N near the roots. See lines 19-24, page 2, where we described the rhizosphere effect. To summarize: 1. the definition of the rhizospheric effect on C and N cycling presented by us coincides with the hypothesis of co-metabolism, which explains the rhizospheric priming effect. Our model is a mathematical description of the phenomenon of co-metabolism inside the rhizosphere, in which microbes uptake carbon and nitrogen from rhizodeposition and causes depolymerization of and metabolises soil organic matter, generating what we call a positive priming effect and net mineralization (so-called N gain – resulting in a positive N balance (net N mineralized and rhizodeposited)). 2. Hence, the model is already consistent with reviwer's comment, e.g. Table 3 for the effect on N supply. 3. We have added a definition of the rhizosphere priming effect.

5. Comment 5: Finally, I think the general reasoning for the set-up of the study could be improved: Why is it importante to derive predictions on rhizosphere C and N cycling for Eucalyptus plantations specifically? Why is rhizosphere priming important?

Answer 5: Eucalyptus plantations occupy an area of approximately 5 million hectares

in Brazil and around 20-25 million hectares in the world. Nitrogen (N) is one of the most accumulated nutrients in Eucalyptus plantations, with values up to 1,300 kg/ha. Interestingly, stands in Brazil often show little or no N fertilization response (DE MELO et al., 2016; PULITO et al., 2015; SMETHURST et al., 2015). The responses, when present, occur within the first two or three years and cease to exist during the time of the harvest (PULITO et al., 2015). Therefore, Eucalyptus plants can meet its high N demand without fertilization or with the application of low doses, between 40 and 70 kg/ha of N, which shows that our understanding of N supply in these systems is incomplete. Within plant types, woody species showed the highest RPE followed by grasses, while crops had the lowest level of the RPE, indicating that plant traits and physiology may exert important controls on the RPE (HUO; LUO; CHENG, 2017). Eucalyptus regnans forests, for example, are able to uptake amounts of N that meet their nutritional requirements and maintain normal photosynthetic activity even in soils with low natural fertility to global standards exposed to fires (DIJKSTRA et al., 2017). The key mechanism that explains this phenomenon is the increase of soil organic matter mineralization induced by root growth (also known as priming effect) (DIJKSTRA et al., 2017). As experimentally proven by Dijkstra et al. (2017), Eucalyptus roots themselves induce greater microbiological activity in soil, and greater respiration and N mineralization. The authors observed two-fold higher microbial biomass and 5.5 higher gross N mineralization in the soil where Eucalyptus plantations grew their roots than in the same soil with root exclusion. As a result, a net reduction of 0.50 kg C m-2 and 12 g N m-2 was observed in the total C and N contents in plots where roots grew in a relatively short period of time (April 2010 to March 2011) (DIJKSTRA et al., 2017).

Experimental evidence for Brazilian sites in the Vale do Rio Doce region (Minas Gerais state) with a hybrid clone of E. grandis x E. urophylla also point to the existence of a link between the root activity and the increase in the rates of mineralization of the forest soil (DELVAUX, 2014). The author observed higher microbial biomass, metabolic quotient (qmic), enzymatic activity (laccase and Mn-peroxidase), potential mineralization and, consequently, lower total N in the rhizosphere than in the bulk soil (DELVAUX, 2014).

In a greenhouse experiment, Hurtarte (2017) observed that the growth of seedlings roots of a hybrid clone of E. grandis x E. urophylla in an oxisol reduced the stocks of C associated mineral fraction in the rhizosphere, especially when plants were under N nutritional limitation.

Therefore, the rhizosphere effect (or rhizosphere priming effect) in eucalypts is of particular interest to the forset plantaiton industry and to the broader scientific community. Despite this, there are still no models to measure the quantitative importance of this process for forest plantations of the genus Eucalyptus. That is why we formed a group of researchers from Brazil and Australia to present this model to Biogeosciences.

6. Comment 6: Specific comments P. 1, l. 11: The unit for fine root length is unusual, please change to m m-2 or similar. Answer: kg/ha, km/ha are units allowed by the international system of units;

P. 1, l. 14: : : :for the prediction of rhizosphere C and N cycling: Accepted and appropriately changed;

P. 1, l. 19: Do you mean root or soil respiration? N mineralization? N immobilization? : : :immigration, and SOM formation. Soil respiration and N mineralization, etc;

P. 1, l. 21: : : :variables that influenced N gain from rhizosphere N mineralization most were: : : Accepted and appropriately changed;

P. 2, l. 6: I think the expectation that additional N is expected in deeper soil layers than in the topsoil is quite surprising: Where should this N come from? N enters the ecosystem from N fixation and N cycling with leaf litter, which are both processes that are mostly occurring in the topsoil. It is the accumulated result of the net N mineralization minus N rhizodeposited (quantity), therefore the relation is proportional. The higher the soil layer, the greater the amount of N gain. I think the reviewer confused concentration with amount, concentration being higher in surface soil.

P. 2, l. 7: : : :higher than: : : : :observation that growth responses: : : Accepted and

appropriately changed;

P. 2, l. 14: : : :rhizosphere processes: : : : : :N supply for some trees: : : Accepted and appropriately changed;

P. 2, l. 16: : : :in the form of dead roots: : : Accepted and appropriately changed;

P.2, l. 32: The abbreviation for the Microbial Carbon and Nitrogen Physiology model is'MCNiP'. Accepted and appropriately changed;

P. 2, l. 33-34: It remains unclear why mineralization rates need to be linked Accepted and appropriately changed. The MCNiP model was not originally presented as part of a commonly used crop produciton model. To do so, we needed to link it to a number of other processes that operate at the broader ecosystem level.

References

BLAGODATSKAYA, E.; KUZYAKOV, Y. Mechanisms of real and apparent priming effects and their dependence on soil microbial biomass and community structure: Critical review. Biology and Fertility of Soils, v. 45, n. 2, p. 115–131, 2008.

CHENG, Weixin et al. Synthesis and modeling perspectives of rhizosphere priming. New Phytologist, v. 201, n. 1, p. 31–44, 2014.

DE MELO, Eduardo Aparecido Sereguin Cabral et al. Responses of clonal eucalypt plantations to N, P and K fertilizer application in different edaphoclimatic conditions. Forests, v. 7, n. 1, p. 1–15, 2016.

DELVAUX, Julio Cesar. Nutrição nitrogenada do eucalipto: regulação rizosférica do suprimento de nitrogênio segundo o modelo do nitrostato. 2014. 112 f. 2014.

DIJKSTRA, Feike A. et al. Enhanced decomposition and nitrogen mineralization sustain rapid growth of Eucalyptus regnans after wildfire. Journal of Ecology, v. 105, n. 1, p. 229–236, 2017.

DIJKSTRA, Feike A. et al. Rhizosphere priming: A nutrient perspective. Frontiers in Microbiology, v. 4, n. JUL, p. 1–8, 2013.

FAGERIA, N.; STONE, L. Physical, chemical, and biological changes in the rhizosphere and nutrient availability. Journal of Plant Nutrition, v. 29, n. 7, p. 1327–1356, 2006.

HILTNER, L. Ueber neuere Erfahrungen und Probleme auf dem Gebiete der Boden-bakteriologie und unter besonderer BerUcksichtigung der Grundungung und Brache. Arb. Deut. Landw. Gesell, v. 98, p. 59–78, 1904.

HUO, Changfu; LUO, Yiqi; CHENG, Weixin. Rhizosphere priming effect: A meta-analysis. 2017. Disponível em: <https://pdfs.semanticscholar.org/bd06/73c5b14308bc4f1d051b793d3a4cef7764b4.pdf>. Acesso em: 13 dez. 2017.

HURTARTE, Luis Carlos. Carbon mineralization in the rhizosphere of Eucalyptus spp. depends on its nitrogen status. 2017. 56 f. Federal University of Viçosa, 2017.

KUZYAKOV, Y.; FRIEDEL, J.K.;; STAHR, K. (artigo) Review of mechanisms and quantification of priming effects.pdf. Soil Biology & Biochemistry, v. 32, p. 1485–1498, 2000.

KUZYAKOV, Y; AL., Et. Photosynthesis controls of rhizosphere respiration and organic mater decomposition . Soil Biology & Biochemistry, v. 33, p. 1915–1925, 2001.

KUZYAKOV, Yakov. Review: Factors affecting rhizosphere priming effects. Journal of Plant Nutrition and Soil Science, v. 165, p. 382–396, 2002.

LÖHNIS, F. NITROGEN AVAILABILITY OF GREEN MANURES. : Soil Science. Soil Science, v. 22, n. 4, p. 253–290, 1926. Disponível em: <http://journals.lww.com/soilsci/citation/1926/10000/nitrogen_availability_of_green_manures.1.aspx>. Acesso em: 17 nov. 2017.

NEWMAN, E. I. The rhizosphere: carbon sources and microbial populations. Ecological interactions in soil: plants, microbes and animals, p. 107–121, 1985. Disponível em: <https://www.cabdirect.org/cabdirect/abstract/19851999297>. Acesso em: 17 nov. 2017.

PAUSCH, Johanna; KUZYAKOV, Yakov. Carbon input by roots into the soil: Quantification of rhizodeposition from root to ecosystem scale. Global Change Biology, n. October, 2017.

PULITO, Ana Paula et al. Available nitrogen and responses to nitrogen fertilizer in brazilian eucalypt plantations on soils of contrasting texture. Forests, v. 6, n. 4, p. 973–991, 2015.

SMETHURST, Philip James et al. Appraisal of the Snap Model for Predicting Nitrogen Mineralization in Tropical Soils Under Eucalyptus. Revista Brasileira de Ciência do Solo, v. 39, n. 2, p. 523–532, 2015. Disponível em: <http://www.scielo.br/scielo.php?script=sci_arttext&pid=S0100-06832015000200523&lng=en&nrm=iso&tlng=en>.

ZHU, Biao et al. Rhizosphere priming effects on soil carbon and nitrogen mineralization. Soil Biology and Biochemistry, v. 76, p. 183–192, 2014. Disponível em: <http://dx.doi.org/10.1016/j.soilbio.2014.04.033>.

Please also note the supplement to this comment:
https://www.biogeosciences-discuss.net/bg-2017-302/bg-2017-302-AC2-supplement.pdf
* * *
**Fig. 1.** Illustration of rhizosphere C and N cycling processes in the ForPRAN model

---

## Referee Comment (RC2) · Anonymous Referee #3 · 26 Jan 2018

The authors apply the MCNiP model to Eucalyptus plantations to estimate the importance of rhizosphere processes to N nutrition in these systems. The inclusion of a plant component to the model is an important development in addition to the other microbial limitations that the authors integrated. The authors use a variety of data sources to validate the model. However, the results presented do little beyond validating the model and there is little discussion of the larger importance of this work. In addition, the main message of this manuscript is unclear given the lack of structure in the paper as well as the numerous language errors throughout.

In addition to the assumptions highlighted by the first reviewer, I am also troubled by the apparent assumption that thicker roots drive a greater rhizosphere stimulation. This assumption is in direct contrast to what was parameterized in the original MCNiP model.

[Figure]

Detailed Comments:

There are numerous language errors and typos throughout the manuscript. The list below does not include all of these errors.

Abstract: Line 13: change "for instance" to including Line 19: Missing and before SOM formation

Introduction: Page 2 Line 1: The authors use i.e. many times to add another clause to the sentence. This should be put in parentheses with (i.e., N mineralization). Or edit these sentences to include it in the sentence structure.

Page 2 Line 7: Replace high with higher.

Page 2 line 14: Rhizosphere is spelled incorrectly.

Page 2 line 19: This value of 1/3 cited for Finzi et al. 2015 is incorrect. In the top 30cm of soils it only approaches 25% when the rhizosphere influence is assumed to be high.

Page 2 line 31: Schimel and Weintraub as well as the Allison reference did not develop the model to look at rhizosphere processes. Also the model is MCNiP not MSNiP. This error is repeated throughout.

Methods:

Page 4 line 18: Cite Finzi et al. 2015 as well

Page 4 Line 26: Replace of with on.

Page 5 Line 34: The second half of this sentence is confusing and unclear.

Page 6 Line 4: The lack of feedback between plant growth and rhizosphere stimulation of N mineralization is key process that is missing in this model.

Page 8 Line 8: "it was used data". Same for Line 18.

Results:

Page 11: This text is unclear but it appears that the model is parametrized to have greater rhizosphere volumes when root diameter is larger. This directly contrasts the assumption in Finzi et al. 2015 that fine low diameter roots are more active and thus have a greater rhizosphere effect.

The results section is mainly validation and does not address key ecological questions nor does it attempt to scale these results up.

Table 1. There are no units for the parameter variables. Same for Table 2.

Figure 7 caption should say kinetic.

Figure 9 caption on instead of in.

Table S1: Why do some variables have dashes instead of values?

Conclusions: These abruptly are presented at the end of the text with little context to gauge whether they were supported. In addition, the discussion does not highlight the importance of the work.
* * *

---

## Referee Comment (RC3) · Anonymous Referee #1 · 1 Feb 2018

The authors present an applied study modelling the soil nitrogen feedbacks to plant rhizodeposition. The subject is relevant to the audience of the journal, as nutrient constraints on SOM stocks and plant growth, especially in tropical soils, have high leverage on our understanding of global CN cycles.

However, I cannot judge the soundness of the paper due to lacking information and confusion of units in the model description.

Without reading the cited publication informations are lacking: - on the evaluation data (what does one data point represent: a different plot, a different treatment, a time series? ...) - lack of information how the the model was calibrated (just the one standard parameterization, or some parameters adjusted, once for all the data or different parameters by dataset or observation, ...)

[Figure]

There are unit errors in the model description E.g. in 21 the units of right and left hand side do not match.

There is confusion between rates (amounts be per hour) and pure amounts (here concentrations per g Soil).

In eq. 16 and 17, it is checked whether a rate is smaller than an amount (Uc < DOC). This makes sense in a model integration using a time step integration of 1hour, but nevertheless is a category mistake. The model description needs to be better separated from the time integration of the model.

It is very hard for the reader to always need to locate all the abbreviations in the tables S1 and S2. I recommend repeating the the abbreviations in the sections, where they are used for the first time. Because of these confusions and abbreviations, I did not check all the equations.

The formulation of uptake (eq. 13, 14) is awkward. The uptake of DOC is proportional to biomass measured in carbon units (BCm) , while the uptake of DON is proportional to biomass measured in nitrogen units. I suggest computing the uptake of N to be stoichiometric with C using the CN-ratio of the DON. If microbial uptake is deliberaty described differential in C and N, I would at least make it proportional to microbial biomass measured in the same units.

N-Immobilization (eq. 19) is computed only by microbial demand. There is no upper limit of the immobilization rate, hence microbial growth is never limited by total N. Is this reasonable assumption for this site?

The transfer of Kpr*DOC(i) (+ Kpr* flux terms) from DOC to SOC needs more justification (Table S3). If the other flux terms were zero, the DOC would quickly go to zero because tranfering all to SOC. I strongly suggest a formulation of the form: DOC(i+1) = DOC(i) + ... and thinking of some alternative to the term: -Kpr DOC(i), e.g. if Kpr limits uptake, I suggest using Kpr in the uptake equation instead.

Table S3: Again to avoid unit confusion, the multiplication with time (1hr) should be noted explicitly, or better a differential formulation (dX/time = input_rate_X - output_rate_X) should be adopted. Please, also report amounts consistently either per gram or per cm3.

It did not become clear to me how the 3-PG estimate of C flux allocated to roots/rhizodeposition is translated to inputs to the soil. Eq. 7 does depend only on root propoerties. If only root properties estimates by 3-PG are used, it should be checked that the sum of rhizodeposition as computed by the full model (eq. 7) is consistent, i.e. equals, the 3-PG C allocation flux to roots.

---

## Referee Comment (RC4) · Alam Khairul (Referee) · 2 Feb 2018

The subject authors addresses here is very important and keeping the broadness of the work and big dataset of the work in mind, I would like to suggest that the manuscript can be published but authors should consider a major revision of their manuscript before they re-submit it in this Journal and elsewhere. Major concerns - The works are mainly based on the data found in Mello et al. (1998), Neves (2000), Leles (2001), Teixeira et al. (2002), Gatto et al. (2003) and Maquere (2008). More similar works could be adopted in the work. -The manuscript is cursorily written. As for example, in the introduction, Page 2, Lines 7, 8, 13, 14, 16 etc. Similar kind of mistakes is there on every page. -The grammar followed in composing the manuscript is not same throughout the manuscript. - The manuscript doesn't read well. - The abbreviations

which have already been defined earlier are not followed throughout the manuscript. -The English must be improved before resubmission or the manuscript can be checked or edited by two native English speakers with similar scientific backgrounds/of the same field. -The sources of the data used for modeling are not written in a recognized format. -Figure presentation must be improved with exact data sources. -For a better understanding of the results and to compare with other works, the authors can divide the results and discussion into two separate sections. Conclusions Page 19 line 4: the word should be "Conclusions". -The authors could not come up with the message of the work. Even, the sentences are not clear in expressing any meaning. All the sentences should be rewritten. SUPPLEMENTARY MATERIAL - Most of the equations used are not presented with the definitions of the components. Minor concerns: Page 1, Line 19, 'and' should be used after the penultimate process. 2.1 Parameter estimation Page 8, Lines 7-8 are not clear. 2.1.2 C and N availability and microbial demand Page 4, Lines 27-28: Sentence meaning should be clearer. Page 6, Line 3: What does it mean by Eucalypt plantation root? Does it mean Eucalyptus plantation roots? 2.2 The evaluation of the rhizospheric model Page 8, Line 18: It is not clear what the authors wanted to say. Figure 4: for giving it self-explained shape, NSE, ME, MAE and RSR should be elaborated again, in the figure title as a legend. Page 11, Figure 6: There are some less-visible numbers. What does it mean? Page 11, Line 12. A full stop is missing. Page 11, Line 22: Three or there? Page 11, Line 30, 32, 35: Check. Sentences are not clear in meaning. Page 13, Line 41; Sentence meaning is not clear. Page 14, 6: the equivalent or 'equivalent'. 2.1 Sensitivity analysis of the ForPRAN model: The subtitle numbering is not correct. Lines 36-37 are not correct in meaning. (Meaning not clear). Table 1. can be revised as "Values of the input variables used in the model to estimate fine root length, rhizosphere volume and C rhizodeposition Figure 2. Flowchart of processes represented in the ForPRAN model: Presentation must be improved. Figure 8 and 9: Source: Based on not based in - In, table 1, soil clay contente should be "Clay content in soil". Page 20, Lines 4-5, Is it the source of the data used for the modeling? Page 21, Line 2, Line 31, the meaning is not

clear. Eq 19, 20, 27, 28 are not well-presented. Page 27, Lines 24-25, the sentences are not clear in meaning.

Please also note the supplement to this comment:
https://www.biogeosciences-discuss.net/bg-2017-302/bg-2017-302-RC4-supplement.pdf

───────────────────────────────

---

## Author Comment (AC3) · 23 Feb 2018

Dear reviewer,

We appreciated your attention and helpful criticisms.

Sincerely yours,

Authors
* * *
Rafael V. Valadares

Soil Department

[Figure]

Universidade Federal de Viçosa

36570-900 Viçosa, MG, Brazil

E-mail: rafaelvvaladares@hotmail.com

Tel: +55 (31) 99253 7720

Answers to anonymous reviewer's comments

1. Comment from Referees: "The authors apply the MCNiP model to Eucalyptus plantations to estimate the importance of rhizosphere processes to N nutrition in these systems. The inclusion of a plant component to the model is an important development in addition to the other microbial limitations that the authors integrated. The authors use a variety of data sources to validate the model. However, the results presented do little beyond validating the model and there is little discussion of the larger importance of this work. In addition, the main message of this manuscript is unclear given the lack of structure in the paper as well as the numerous language errors throughout."

Author's response: Once more, we thank the gentle comments regarding our work. In fact, at the end of the work, we presented the question of the biological importance. We showed this question since the beginning of the paper, in the abstract and introduction, for instance. This question is important considering its environmental impact and also justifies part of our study. That is the main reason to present it in the same paper of the model description. It was also one of the main motivations for elaborating this model for genus Eucalyptus. About the structure, we presented a theoretical and mathematical model with its validation, sensitivity analysis, and biological importance. It is a very common structure in this kind of work. Many thanks for this comment.

Author's changes in the manuscript: We will submit the paper to a rigorous language review.

2. Comment from Referees:"In addition to the assumptions highlighted by the first reviewer, I am also troubled by the apparent assumption that thicker roots drive a greater

rhizosphere stimulation. This assumption is in direct contrast to what was parameterized in the original MCNiP model."

Author's response: Our model uses the root diameter to simulate the rhizosphere process considering the percentage of root length ratio per diameter. Around 88 % of the root length belong to those roots with a diameter lower than 1 mm (graphic available in the attached version). Thus, if we divide the rhizosphere N supply output, the highest supply still comes from roots with a diameter lower than 1 mm. This assumption was done with base in relevant papers about Eucalyptus roots, cited in the present paper (Baldwin and Stewart, 1987; Mello et al., 1998). Therefore, the highest contribution to the rhizosphere process comes from roots with diameter <= 1 mm because of the higher length and, consequently, rhizosphere volume and rhizodeposition flux. Thus, when we say in the sensitivity analysis that there is a higher supply until 3 mm, we are just doing a simple sum of the contribution of the rhizosphere from the root classes of 0-1 mm plus 1-2 mm and 2-3 mm - the total supply of the active rhizosphere system.

Author's changes in the manuscript: None.

3. Comment from Referees: "There are numerous language errors and typos throughout the manuscript. The list below does not include all of these errors. Abstract: Line 13: change "for instance" to including Line 19: Missing and before SOM formation."

Author's response: About the line 13, in this case, "for instance" was used as an example, which is correct. About the line 19, it is not clear the sense of the suggestion.

Author's changes in the manuscript: None.

4. Comment from Referees: Page 2 Line 1: The authors use i.e. many times to add another clause to the sentence. This should be put in parentheses with (i.e., N mineralization). Or edit these sentences to include it in the sentence structure.

Author's response: Thanks. The correction will be made in the final version.

Author's changes in the manuscript: The correction will be made in the final version.

5. Comment from Referees: "Page 2 Line 7: Replace high with higher."

Author's response: The correction will be made in the final version.

Author's changes in the manuscript: The correction will be made in the final version.

6. Comment from Referees: "Page 2 line 14: Rhizosphere is spelled incorrectly."

Author's response: The correction will be made in the final version.

Author's changes in the manuscript: The correction will be made in the final version.

7. Comment from Referees: "Page 2 line 19: This value of 1/3 cited for Finzi et al. 2015 is incorrect. In the top 30cm of soils, it only approaches 25% when the rhizosphere influence is assumed to be high." Author's response: We were based on the following sentence from Finzi et al. (2015) abstract: "Then, using a numerical model that combines rhizosphere effect sizes with fine root morphology and depth distributions, we show that root-accelerated mineralization and priming can account for up to one-third of the total C and N mineralized in temperate forest soils."

Author's response: After your suggestion, we read the paper again and also saw that the value of 25 % is the correct. Thanks for the valuable suggestion. We will correct it in the final version. It's somehow interesting because we are preparing other paper with scenarios simulation and this value (proportion) it is very close to those from some of four main forests plantations.

Author's changes in the manuscript: The correction will be made in the final version.

8. Comment from Referees: Page 2 line 31: Schimel and Weintraub, as well as the Allison reference, did not develop the model to look at rhizosphere processes. Also, the model is MCNiP not MSNiP. This error is repeated throughout.

Author's response: Since we are developing an organism-oriented model, it is not a problem to mention the advances of these authors. The main changes, in terms of representation, is the availability of growth resources for microorganisms, when we

compare the bulk soil with rhizosphere soil. Thus, advances are, in a sense, shared for the representation of any subsystem of the whole soil system. In mechanistic models, it is always good to take into account the phenomena in question in the highest degree of importance. Regarding the name of the model, in fact, during the review process, we substituted the letter C for S. Thank you for the valuable observation.

Author's changes in the manuscript: The name MCNiP will be corrected in the final version.

9. Comment from Referees: Page 4 line 18: Cite Finzi et al. 2015 as well

Author's response: The citation will be included in the final version.

Author's changes in the manuscript: The citation will be included in the final version.

10. Comment from Referees: Page 4 Line 26: Replace of with on.

Author's response: It will be corrected in the final version of the article.

Author's changes in the manuscript: It will be corrected in the final version of the article.

11. Comment from Referees: Page 5 Line 34: The second half of this sentence is confusing and unclear.

Author's response: It will be improved to the final version of the article.

Author's changes in the manuscript: It will be improved to the final version of the article.

12. Comment from Referees: Page 6 Line 4: The lack of feedback between plant growth and rhizosphere stimulation of N mineralization is key process that is missing in this model.

Author's response: This type of refinement is timely, but it should be the focus of an article focused on growth modeling. Our team is already working on it.

Author's changes in the manuscript: None.

13. Comment from Referees: Page 8 Line 8: "it was used data". Same for Line 18.

Author's response: It will be corrected in the final version of the article.

Author's changes in the manuscript: It will be corrected in the final version of the article.

14. Comment from Referees: Page 11: This text is unclear but it appears that the model is parametrized to have greater rhizosphere volumes when root diameter is larger. This directly contrasts the assumption in Finzi et al. 2015 that fine low diameter roots are more active and thus have a greater rhizosphere effect.

Author's response: Our model uses the root diameter to simulate the rhizosphere process considering the percentage of root length ratio per diameter. Around 88 % of the root length belong to those roots with a diameter lower than 1 mm (graphic available in the attached version). Thus, if we divide the rhizosphere N supply output, the highest supply still comes from roots with a diameter lower than 1 mm. This assumption was done with base in relevant papers about Eucalyptus roots, cited in the present paper (Baldwin and Stewart, 1987; Mello et al., 1998). Therefore, the highest contribution to the rhizosphere process comes from roots with diameter <= 1 mm because of the higher length and, consequently, rhizosphere volume and rhizodeposition flux. Thus, when we say in the sensitivity analysis that there is a higher supply until 3 mm, we are just doing a simple sum of the contribution of the rhizosphere from the root classes of 0-1 mm plus 1-2 mm and 2-3 mm - the total supply of the active rhizosphere system.

Author's changes in the manuscript: The results are consistent with data from field experiments.

15. Comment from Referees: The results section is mainly validation and does not address key ecological questions nor does it attempt to scale these results up.

Author's response: We thank the comment. But the reviewer should consider that the fundamental question is the priming effect and the mathematical representation of the rhizosphere system with respect to C and N. We did it! And, more importantly, we

consider biological logic by representing the effect of the availability of biotic and abiotic growth factors. This question is the basis of all work and was, within the possibilities, represented.

We have reviewed numerous articles on this subject and are aware of our contribution. We gave ecosystem scale when considering the plant in this system, which we have seen to be rare in this type of work. In addition, we devote an entire section to the biological importance of this phenomenon.

Author's changes in the manuscript: None.

16. Comment from Referees: Table 1. There are no units for the parameter variables. Same for Table 2.

Author's response: Thanks. We will correct it in the final version.

Author's changes in the manuscript: Thanks. We will correct it in the final version.

17. Comment from Referees: Figure 7 caption should say kinetic.

Author's response: Thanks. We will correct it in the final version.

Author's changes in the manuscript: Thanks. We will correct it in the final version.

18. Comment from Referees: Figure 9 caption on instead of in.

Author's response: Thanks. We will correct it in the final version.

Author's changes in the manuscript: We will correct it in the final version.

19. Comment from Referees: Table S1: Why do some variables have dashes instead of values?

Author's response: Some variables are user-defined and others are model outputs.

Author's changes in the manuscript: none.

20. Comment from Referees: Conclusions: These abruptly are presented at the end

of the text with little context to gauge whether they were supported. In addition, the discussion does not highlight the importance of the work.

Author's response: The proposition of this model in itself justifies the article and is already helping in the understanding of results of measurements of SOC in Brazil, with a view to assisting in the management of forest residues and nitrogen fertilization. Discussions about scenario simulation are quite interesting and will be covered in another article. In any case, scenario simulations should not overshadow the proposition of the model that gives rise to them. Finally, the presentation of biological importance was proposed as a means of showing the reader the direct importance of the work.

Author's changes in the manuscript: none.

—————————————————————

[Figure]

[Figure]

**Fig. 1.** Percentage of the total root length per root diameter

---

## Author Comment (AC4) · 27 Feb 2018

Dear reviewer,

We appreciate the anonymous reviewer's attention to detail and the helpful comments.

Sincerely yours,

Rafael V. Valadares

–

Rafael Vasconcelos Valadares

Ph.D. in Soils and Plant Nutrition/UFV

[Figure]

Laboratory of Microbial Ecology

+55 031 99253 772

Answers to anonymous reviewer's comments

Comment from Referees:" However, I cannot judge the soundness of the paper due to lacking information and confusion of units in the model description"

Author's response: Our model is running using the processes explained in the paper. Furthermore, the model is an improvement and specific application of a previously published model MCNiP – article entitled: "Stoichiometry constrains microbial response to root exudation-insights from a model and a field experiment in a temperate forest.

Some values are coeficients and have no unit.

Author's changes in the manuscript: We will review the units.

Comment from Referees: Without reading the cited publication informations are lacking: - on the evaluation data (what does one data point represent: a different plot, a different treatment, a time series? ...) - lack of information how the model was calibrated (just the one standard parameterization, or some parameters adjusted, once for all the data or different parameters by dataset or observation, ...)

Author's response: We wrote a topic inside the Material and Methods about the parameterization.

Author's changes in the manuscript: None.

Comment from Referees: There are unit errors in the model description E.g. in 21 the units of right and left hand side do not match.

Author's response:

EPc = Kep BCm

EPc = rate of enzymes production ($\mu$g/g/h)

Rate of enzymatic production per unit of biomass (Kep) = $\mu$g/$\mu$g/h

C in microbial biomass (BCm) = $\mu$g/g

EPc = ($\mu$g/$\mu$g/h)x $\mu$g/g

EPc= $\mu$g/g/h

It is correct. This means that the rate of the enzymes production per gram of soil in one step of the model.

Author's changes in the manuscript: Thank you for presenting your opinion.

Comment from Referees: There is confusion between rates (amounts be per hour) and pure amounts (here concentrations per g Soil).

Author's response: All the concentrations are with the correct unit.

Since our time step is one hour, we consider that it is unnecessary to express time (1 h) in all the rates. But we will change it.

Author's changes in the manuscript: We will insert explicitly the unit of the hour in all the rates. Many thanks for this suggestion.

Comment from Referees: In eq. 16 and 17, it is checked whether a rate is smaller than an amount (Uc < DOC). This makes sense in a model integration using a time step integration of 1 hour, but nevertheless is a category mistake. The model description needs to be better separated from the time integration of the model.

Author's response: It just means that the microbiota may only depolymerize smaller amounts of C and N from soil organic matter than it can take up. The enzymes compete for substrate sites and the soil itself competes for the substrate cleaved by them. Then, the amount of dissolved carbon and nitrogen clearly may be less than the uptake capacity of the microbiota.

Author's changes in the manuscript: Thank you for submitting your opinion.

Comment from Referees: It is very hard for the reader to always need to locate all the abbreviations in the tables S1 and S2. I recommend repeating the the abbreviations in the sections, where they are used for the first time. Because of these confusions and abbreviations, I did not check all the equations.

Author's response: If the manuscript goes on to later steps, such changes will be made.

Author's changes in the manuscript: We will repeat the abbreviations in the text.

Comment from Referees: The formulation of uptake (eq. 13, 14) is awkward. The uptake of DOC is proportional to biomass measured in carbon units (BCm) , while the uptake of DON is proportional to biomass measured in nitrogen units. I suggest computing the uptake of N to be stoichiometric with C using the CN-ratio of the DON. If microbial uptake is deliberaty described differential in C and N, I would at least make it proportional to microbial biomass measured in the same units.

Author's response:

Uc=(Vmaxuptake BCm DOC)/(Kmuptake+DOC)

Un=(Vmaxuptake BNm DON)/(Kmuptake+DON)

It already considers the dissolved C and N stoichiometric. DOC means organic C in solution. DON means organic N in solution.

DOC (i+1)=(1- Kpr)(DOC(i)+Ce+Dc+CYc+ELc)-Uc

DON (i+1)=(1- Kpr)(DON(i)+Ne+Dn+CYn+ ELn)-Un

Author's changes in the manuscript: Thank you for submitting your opinion.

Comment from Referees: N-Immobilization (eq. 19) is computed only by microbial demand. There is no upper limit of the immobilization rate, hence microbial growth is never limited by total N. Is this reasonable assumption for this site?

Author's response:

[Figure]

The amount of dissolved carbon absorbed limits the amount of N immobilization (most of the time, microbes are carbon limited!) (eq. 19, SUPPLEMENTARY MATERIAL). As the absorbed carbon is not unlimited, which is explained by the Michaelis Menten equation, C uptake represents the main upper limit (eq. 19). Respiration also limits N immobilization, as well as the carbon respiration for the enzymes production, for example. But if there is no inorganic N, immobilization also is limited. In this case, carbon overflow metabolism is activated to keep the microbial C/N ratio constant (eq. 28).

Author's changes in the manuscript: Thank you for submitting your opinion.

Comment from Referees: The transfer of Kpr*DOC(i) (+ Kpr* flux terms) from DOC to SOC needs more justification (Table S3). If the other flux terms were zero, the DOC would quickly go to zero because tranfering all to SOC. I strongly suggest a formulation of the form: DOC(i+1) = DOC(i) + ... and thinking of some alternative to the term: -Kpr DOC(i), e.g. if Kpr limits uptake, I suggest using Kpr in the uptake equation instead.

Author's response: Kpr is the effect of soil competing with microbes by the dissolved organic carbon and nitrogen. If the fluxes of DOC and DON reach zero, the product is zero. Thus, the soil formation will be zero in the rhizosphere soil. Some improvements in this part of work still depend on experimental evidence about C and N saturation on the rhizosphere soil. In the future, it can be established a relationship between mineralogy/texture, carbon, and nitrogen deficit and soil protection (Kpr).

Author's changes in the manuscript: Thank you for submitting your opinion.

Comment from Referees: Table S3: Again to avoid unit confusion, the multiplication with time (1hr) should be noted explicitly, or better a differential formulation (dX/time = input_rate_X - output_rate_X) should be adopted. Please, also report amounts consistently either per gram or per cm3

Author's response: We will consider rates with the unit plus hour-1 explicitly. The time

step of the model is one hour, that is why it was not previously considered.

Author's changes in the manuscript: We will consider rates with the unit plus hour-1 explicitly.

Comment from Referees: It did not become clear to me how the 3-PG estimate of C flux allocated to roots/rhizodeposition is translated to inputs to the soil. Eq. 7 does depend only on root propoerties. If only root properties estimates by 3-PG are used, it should be checked that the sum of rhizodeposition as computed by the full model (eq. 7) is consistent, i.e. equals, the 3-PG C allocation flux to roots.

Author's response: 3-PG does not consider it explicitly. That is why we presented equations 1 to 8 (SUPPLEMENTARY MATERIAL).

The values of rhizodeposition are consistent with measurements done for trees.

Author's changes in the manuscript: Thank you for submitting your opinion.

---

## Author Comment (AC5) · 27 Feb 2018

Dear Dr. Alam Khairul,

We appreciate your attention to detail and the helpful comments.

Sincerely yours,

Rafael V. Valadares
* * *
Ph.D. in Soils and Plant Nutrition/UFV

Laboratory of Microbial Ecology

[Figure]

+55 031 99253 772

Answers to reviewer's comments

Major Comments

1. Comment from Referees: The subject authors addresses here is very important and keeping the broadness of the work and big dataset of the work in mind, I would like to suggest that the manuscript can be published but authors should consider a major revision of their manuscript before they re-submit it in this Journal and elsewhere. Major concerns - The works are mainly based on the data found in Mello et al. (1998), Neves (2000), Leles (2001), Teixeira et al. (2002), Gatto et al. (2003) and Maquere (2008).

Author's response: Thanks for the comment, but it is not clear what the major concern is or what suggested changes here might be. These datasets were used in the first part of the model (root growth). Other references were used later during modeling of C and N cycling in the rhizosphere.

Changes to the manuscript: We will include other datasets in the root growth model, considering the subsequent steps.

2. Comment from Referees: The manuscript is cursorily written. As for example, in the introduction, Page 2, Lines 7, 8, 13, 14, 16 etc. Similar kind of mistakes is there on every page. -The grammar followed in composing the manuscript is not same throughout the manuscript. - The manuscript doesn't read well. - The abbreviations which have already been defined earlier are not followed throughout the manuscript. - The English must be improved before resubmission or the manuscript can be checked or edited by two native English speakers with similar scientific backgrounds/of the same field.

Author's response: We appreciate this review. We have made major corrections to the English, trying to make the text more fluid.

Changes to the manuscript: We will submit the work to a specialized English review

service.

3. Comment from Referees: The sources of the data used for modeling are not written in a recognized format.

Author's response: We will include in the tables of supplementary material S1 and S2 a data source column. Currently, the references are at the bottom part of these tables.

Author's changes in the manuscript: We will adjust the tables S1 and S2.

4. Comment from Referees: Figure presentation must be improved with exact data sources

Author's response: Some extra explanation can be provided in the subtitle.

Author's changes in the manuscript: We have improved this aspect of figure presentation.

5. Comment from Referees: For a better understanding of the results and to compare with other works, the authors can divide the results and discussion into two separate sections.

Author's response: We presented the work with results and discussion together because the article already has a large number of pages.

Author's changes in the manuscript: None

6. Comment from Referees: Conclusions Page 19 line 4: the word should be "Conclusions". -The authors could not come up with the message of the work. Even, the sentences are not clear in expressing any meaning. All the sentences should be rewritten.

Author's response: These sentences will be improved.

Author's changes in the manuscript: We will improve this topic.

7. Comment from Referees: Most of the equations used are not presented with the

definitions of the components

Author's response: Thank you, but we disagree. All the equations with definitions are in the supplementary material.

Author's changes in the manuscript: None.

Minor Comments

8. Comment from Referees: Minor concerns: Page 1, Line 19, 'and' should be used after the penultimate process. 2.1 Parameter estimation Page 8,

Author's response: All possible errors related to language aspects have been corrected with the assistance of a specialized professional.

Author's changes in the manuscript: It will be properly corrected in the final version.

9. Comment from Referees: Lines 7-8 are not clear. 2.1.2 C and N availability and microbial demand Page 4, Lines 27-28: Sentence meaning should be clearer.

Author's response: We are making improvements in the mentioned sections in order to make them clearer.

Author's changes in the manuscript: We are making improvements in the mentioned sections in order to make them clearer.

10. Comment from Referees: Page 6, Line 3:What does it mean by Eucalypt plantation root? Does it mean Eucalyptus plantation roots?

Author's response: Sorry. The model simulates the root growth dynamics.

Author's changes in the manuscript: It means eucalypts roots.

11. Comment from Referees: 2.2 The evaluation of the rhizospheric model Page 8, Line 18: It is not clear what the authors wanted to say.

Author's response: We used the model to simulate real conditions in which we have

field measurements from literature data. Then, we fit a linear model of type y = b1x + b0, having the value estimated by the model in the X-axis and the value observed by field experiments in the Y-axis (literature data). We evaluated the performance of the model through the coefficient of determination ($R^2$). In addition, we tested the coefficients b1 considering the null hypothesis equal to 1; and the coefficient b0 considering the null hypothesis equal to 0. An ideal model must have b1 not different from 1 and b0 not different from 0. We used the t-test to evaluate these hypotheses.

Author's changes in the manuscript: We will provide additional explanations.

12. Comment from Referees: Figure 4: for giving it self-explained shape, NSE, ME, MAE and RSR should be elaborated again, in the figure title as a legend. Page 11, Figure 6: There are some less-visible numbers. What does it mean? Page 11, Line 12. A full stop is missing.

Author's response: If these are defined in the methods section, then I don't think we need to write them again in each figure where used. Figure 6 is a qualitative assessment. The figure will be edited to show no trace of numbers. Corrections will be made on line 12.

Author's changes in the manuscript: We will edit Figure 6 and make corrections on line 12.

13. Comment from Referees: Page 11, Line 22: Three or there? Page 11, Line 30, 32, 35: Check. Sentences are not clear in meaning. Page 13, Line 41; Sentence meaning is not clear. Page 14, 6: the equivalent or 'equivalent'.

Author's response: Page 11, line 22: There. Page 11, line 30, 32, 35: We will improve the sentences mentioned.

Author's changes in the manuscript: We will improve the sentences mentioned in the final version.

14. Comment from Referees: 2.1 Sensitivity analysis of the ForPRAN model: The

subtitle numbering is not correct. Lines 36-37 are not correct in meaning. (Meaning not clear). Table 1. can be revised as "Values of the input variables used in the model to estimate fine root length, rhizosphere volume and C rhizodeposition

Author's response: Subtitle numbering is correct. Lines 36-37: We will clarify. Table 1.: The values are correct.

Author's changes in the manuscript: Lines 36 and 37 will be reformulated.

15. Comment from Referees: Figure 2. Flowchart of processes represented in the ForPRAN model: Presentation must be improved. Figure 8 and 9: Source: Based on not based in - In, table 1, soil clay contente should be "Clay content in soil".

Author's response: The flowchart seems easy to understand. Anyway, we will evaluate improvements. Figure 8 and 9 and table 1: We will correct these details.

Author's changes in the manuscript: Figure 8 and 9 and table 1: We will correct these details.

16. Comment from Referees: Page 20, Lines 4-5, Is it the source of the data used for the modeling? Page 21, Line 2, Line 31, the meaning is not clear. Eq 19, 20, 27, 28 are not well-presented. Page 27, Lines 24-25, the sentences are not clear in meaning.

Author's response: Lines 4-5: Yes, they are the sources used for the elaboration of this part of the work. Page 21, Line 2, Line 31, Eq 19, 20, 27, 28; Page 27, Lines 24-25: We will improve these parts.

Author's changes in the manuscript: We will improve the less clear parts mentioned considering the next step of the publication process.

---

## Author Comment (AC6) · 27 Feb 2018

Dear Dr. Jens-Arne Subke,

Consider the attached responses to the reviewers' comments, which were addressed to the article entitled "Modeling rhizosphere carbon and nitrogen cycling in Eucalyptus plantation soil"(previous Manuscript ID: 412375), authored by Rafael V. Valadares, Júlio C. L. Neves, Maurício D. Costa, Philip J. Smethurst, Luiz A. Peternelli, Guilherme L. de Jesus, Reinaldo B. Cantarutti, and Ivo R. da Silva, to be considered for publication in Biogeosciences. First of all, and on behalf of the co-authors, I would like to thank you again and the reviewers for all the suggestions made to improve our manuscript. We believe that if our paper pass to the next step, the new version will address most

of the reviewers' comments, including improvements in all article topics. We take this opportunity to say that we were very careful with each reviewer, trying to clarify the points which raised doubts. Thus, I believe that the objectives of the discussion stage were achieved to give directions for an improved new version. A detailed point-by-point reply was presented in the answer sheets for each reviewer. We hope that, in the next step of the publication process, the manuscript will meet all quality requirements for publication in Biogeosciences. Once again, we are very thankful for your editorial work and the opportunity of exposing our work. Additionally, the manuscript has not been submitted for publication elsewhere and accordingly all authors contributed to the work described here and take entire responsibility for it. There is also no conflict of interest.

Sincerely yours,

Rafael V. Valadares

PhD in Soils and Plant Nutrition

Laboratory of Microbial Ecology

Universidade Federal de Viçosa

36570-900 Viçosa, MG, Brazil

E-mail: rafaelvvaladares@hotmail.com

Tel: +55 (31) 99253 7720

---

## Author Response (AR1)

Viçosa, April 27th, 2018

Dear Dr. Jens-Arne Subke

Thank you for your comments of 23rd March 2018 on our responses to reviewers comments on the article entitled "**Modeling rhizosphere carbon and nitrogen cycling in *Eucalyptus* plantation soil**" (Manuscript bg-2017-302), authored by Rafael V. Valadares, Júlio C. L. Neves, Maurício D. Costa, Philip J. Smethurst, Luiz A. Peternelli, Guilherme L. de Jesus, Reinaldo B. Cantarutti, and Ivo R. da Silva, to be considered for publication in the journal **Biogeosciences**.

First of all, and on behalf of the co-authors, I thank you again and the reviewers for all suggestions. We believe that our responses below and the new version of the manuscript address the most important points of the reviewer's comments. In some cases, suggested changes were not made and justifications are provided in our specific responses to each reviewer in a detailed point-by-point response presented below. We hope the manuscript now meets all quality requirements for publication in **Biogeosciences**.

Once again, we are very thankful for your editorial work and the improvements that it has prompted. Additionally, the manuscript has not been submitted for publication elsewhere and accordingly all authors contributed to the work described here and take entire responsibility for it. There is also no conflict of interest.

Sincerely yours,

Rafael V. Valadares
Departamento de Solos
Universidade Federal de Viçosa
36570-900 Viçosa, MG, Brazil
E-mail: rafaelvvaladares@hotmail.com
Tel:  +55 (31) 99253 7720

**Response to Comments by Associate Editor**

**Comment from the Associate Editor:** You manage to defend some of the criticism reasonably, but there are several instances where I think you fail to appreciate the referees' points, and you have to adapt your text to accommodate concerns.

**Authors' response:** Accepted. We revisited all of our previous responses and made numerous changes to the manuscript. These changes are described in detail below for each reviewer report.

**Comment from the Associate Editor:** For example, referee #2's point regarding an apparent assumption that larger diameter roots have more influence on rhizosphere effects highlights an area where your presentation was not sufficiently clear. Rather than correcting the referee and insisting that your point is correct, you should make sure that the way you present C allocation in relation to root diameter is absolutely clear to readers.

**Comment from Referee #3 in Report #1:** " . . . I am also troubled by the apparent assumption that thicker roots drive a greater rhizosphere stimulation. This assumption is in direct contrast to what was parameterized in the original MCNiP model."

**Authors' response:** Our explanation might have been misleading because of some incorrect use of English. We reassure you that the ForPRAN model does not assume that larger roots have more influence on rhizosphere process. Thus, of total simulated rhizosphere N supply, the highest portion comes from roots of the smallest class, i.e. with a diameter less than 1 mm. This assumption was based in relevant papers about *Eucalyptus* roots, cited in the present paper (Baldwin and Stewart, 1987; Mello et al., 1998). Our description of 'root diameter' in the sensitivity analysis was probably the cause of this confusion, as it was actually 'root diameter maximum for fine roots', which is now how it is labelled in Tables 1 and 2. As this upper boundary increases for considering what is a fine root, total root length assumed to affect rhizosphere processes also increases, and hence total rhizosphere N supply is also very sensitive to this parameter.

**Comment from the Associate Editor:** The same applies to many of your responses, where you should make sure that any misunderstanding highlighted by a referee comment is not a fault of the referee, but of the clarity of your presentation.

**Authors' response:** Accepted. As mentioned above, we revisted all of our previous responses and made numerous changes to the manuscript. These changes are described in detail below for each reviewer report, which we believe has improved clarity.

**Comment from the Associate Editor:** The discussion needs to capture more of the criticism voiced by referees, rather than just review findings. You are clear about the fact that the model does not include feedback between priming and plant growth. This would be desirable, but was not art of your approach – it should be discussed as an important current gap and area for improvement in future research.

**Authors' response:** Accepted. The discussion has been improved along these lines.

**Comment from the Associate Editor:** In your conclusion, please present a continuously phrased paragraph, rather than bullet points. Instead of simply repeating results, your conclusions should state clearly what the key outcome of your presented research is, and how it advances our understanding.

**Authors' response:** Accepted and changes made.

**Comment from the Associate Editor:** There are a couple of specific comments below I picked up from referee comments and your response to them below. Overall, I think that substantial revision is required for this manuscript to be ready for publication.

**Authors' response:** Accepted – substantial revision has been made.

**Comment from the Associate Editor:** P. 1, Line 13: Please replace "for instance" by "including", as suggested by refere #2 P. 1, Line 19: Add "and" before "SOM formation", as suggested by referee #3

**Authors' response:** Accepted and change made.

**Comment from the Associate Editor:** The sentence clearly suggests that the authors mentioned developed the MCNiP model to specifically investigate rhizospheric N cycling (comment referee #3). If this is not actually the case, then this has to be re-phrased to reflect what precisely these authors did to develop the model.

**Authors' response:** Accepted and change made.

**Response to Comments by Reviewers in Addition to Those Provided Earlier**

We present here additional responses to those provided earlier.

If we thought an earlier comment was adequate, and it wasn't specifically raised by the associate editor, it is not repeated here. If the associate editor or reviewers think we should revisit any of these comments, we will be pleased to do so.

**Reviewer Report #2 (Referee #1):**

**Comment from Referees:** (...) numerous language errors throughout

**Authors' changes in the manuscript:** The manuscript was rechecked and edited to improve English expression by one of our co-authors (P. Smethurst) whose first language is English.

**Comment from Referees:** Page 2 Line 1: The authors use i.e. many times to add another clause to the sentence. This should be put in parentheses with (i.e., N mineralization). Or edit these sentences to include it in the sentence structure.

**Authors' response:** Accepted that there was overuse of 'i.e.' and we have appropriately changed it in several places. However, we kept some useage of 'i.e.' consistent with common useage in the scientific literature.

**Text changed:** , e.g. at this location (...) from decomposition of soil organic matter (i.e., N mineralization) (...).

**Comment from Referees:** "Page 2 Line 7: Replace high with higher."

**Authors' response:** Accepted and appropriately changed;

**Text changed:** "These rates of N supply were similar to or higher than the N demand (…)"

**Comment from Referees:** "Page 2 line 14: Rhizosphere is spelled incorrectly."

**Authors' response:** Accepted and appropriately changed;

**Text changed:** There is speculation that rhizosphere processes might be a significant source of N supply for some trees (…).

**Comment from Referees:** "Page 2 line 19: This value of 1/3 cited for Finzi et al. 2015 is incorrect. In the top 30 cm of soils, it only approaches 25 % when the rhizosphere influence is assumed to be high."

**Authors' response:** Accepted and appropriately changed;

**Text changed:** Finzi et al. (2015) estimated that the mineralization in rhizosphere soil of temperate forests can represent 1/4 of all mineralized N in the ecosystem.

**Comment from Referees:** (...) Also, the model is MCNiP not MSNiP.
**Authors' response**: Accepted and appropriately changed;

**Comment from Referees:** Page 4 line 18: Cite Finzi et al. 2015 as well
**Authors' response:** Accepted and appropriately changed;

**Text changed:** The model follows the logic of stoichiometric balance between substrate supply and microbial demand and the 'Law of the Minimum' applied to the microbial processes, as presented by Schimel and Weintraub (2003), Allison et al. (2010), Drake et al. (2013) and Finzi et al. (2015).

**Comment from Referees:** Page 4 Line 26: Replace of with on.
**Authors' response:** Accepted and appropriately changed;

**Text changed:** (…), mineralogy and on soil carbon content (degree of saturation of clays by organic carbon) (…).

**Comment from Referees:** Page 5 Line 34: The second half of this sentence is confusing and unclear.

**Authors' response:** Accepted and appropriately changed;

**Text changed:** To do so, we modified the MCNiP (Schimel and Weintraub, 2003; Allison et al., 2010; Drake et al., 2013), in order to consider the effect of soil moisture, physical conditions, temperature (effect on exo-enzymes kinetics), immigration and microbial emigration on microbial population dynamics (figure 2).

**Comment from Referees:** Page 8 Line 8: "it was used data". Same for Line 18.
**Authors' response:** Accepted and appropriately changed;

**Text changed:** In addition, to the parameters used in those studies, it was used data presented (...); (...)It was used a linear regression of predicted (P) microbial biomass vs observed (O) data and its coefficient of determination ($R^2$), being tested $\beta 0$ (H0=0) and $\beta 1$(H0=1).

**Comment from Referees:** Table 1. There are no units for the parameter variables. Same for Table 2.

**Authors' response:** Accepted and appropriately changed;

**Comment from Referees:** Figure 7 caption should say kinetic.
**Authors' response:** Accepted and appropriately changed;

**Text changed:** Figure 7. Effect of temperature on enzymatic kinetic represented by KappaD.

**Comment from Referees:** Figure 9 caption on instead of in.
**Authors' response:** Accepted and appropriately changed;

**Text changed:** Figure 9. Effect of soil moisture on relative microbial biomass

**Reviewer Report #3 (Referee #2):**

**Comment from Referees:**" However, I cannot judge the soundness of the paper due to lacking information and confusion of units in the model description"

**Authors' response:** Text changed.  We believe we have generally improved the information available and clarity of its presentation.  For example, (a) . . . and (b) . . . . .????

Units were revised and "hour" used in rates. Since we had already noted that the time step of the model is 1 hour, this unit had been previously omitted.

**Comment from Referees:** Without reading the cited publication informations are lacking:
- on the evaluation data (what does one data point represent: a different plot, a different treatment, a time series? ...) - lack of information how the model was calibrated (just the one standard parameterization, or some parameters adjusted, once for all the data or different parameters by dataset or observation, ...)

**Authors' response:** We wrote a topic inside the Material and Methods about the parameterization. In addition, we present in the material and methods tables of supplementary material with numbers that are associated with their references.
**Authors' changes in the manuscript:** None.

**Comment from Referees:** It is very hard for the reader to always need to locate all the abbreviations in the tables S1 and S2. I recommend repeating the the abbreviations in the sections, where they are used for the first time.

**Authors' response:** We understand the Authors' comment, but have decided not to follow the recommendation as it will considerably increase the length of the manuscript, and we believe that an interested reader will probably not mind deciphering the equations using the tables of abbreviations. The equations are ordered by numbers and the symbols are present in the tables (Tables S1 and S2). However, we leave it to the editor to decide whether or not to have us follow the recommendation to define the symbols in the text as well as in the tables.

**Reviewer Report #4 (Referee #3):**

**Comment from Referees:** P. 1, l. 14: : : :for the prediction of rhizosphere C and N cycling:

**Authors' response:** Accepted and appropriately changed;

**Text changed:** "Therefore, the objective of this work was to propose and evaluate a mechanistic model for the prediction of rhizosphere C and N cycling in *Eucalyptus* plantations."

**Comment from Referees:** P. 1, l. 19: Do you mean root or soil respiration? N mineralization? N immobilization? immigration, and SOM formation. Soil respiration and N mineralization, etc;

**Authors' response:** Accepted and appropriately changed;

**Text changed:** "(…) N mineralization, N immobilization, microbial death, microbial emigration and immigration, SOM formation (…)"

**Comment from Referees:** P. 1, l. 21: : : :variables that influenced N gain from rhizosphere N mineralization most were: : :

**Authors' response:** Accepted and appropriately changed;

**Text changed:** "The input variables that influenced N gain from rhizosphere N mineralization most were (in order of decreasing importance): root diameter > rhizosphere thickness > soil temperature > clay content."

**Comment from Referees:** P. 2, l. 7: : : :higher than: : : : :observation that growth responses: : :

**Authors' response:** Accepted and appropriately changed;

**Text changed:** "These rates of N supply were similar to or higher than the N demand of young plantations in the region, and therefore consistent with the observation that growth responses to N fertilization were minor or absent."

**Comment from Referees:** P. 2, l. 14: : : :rhizosphere processes: : : : : :N supply for some trees: : :

**Authors' response:** Accepted and appropriately changed

**Text changed:** "There is speculation that rhizosphere processes might be a significant source of N supply for some trees (Grayston et al., 1997), (…)"

**Comment from Referees:** P. 2, l. 16: : : :in the form of dead roots: : :

**Authors' response:** Accepted and appropriately changed; **Text changed**: "This effect is mainly due to the release of carbon to the soil in the form of dead roots or rhizodepositions (secretions, lysates, gases, mucilages, etc)."

**Comment from Referees:** P.2, l. 32: The abbreviation for the Microbial Carbon and Nitrogen Physiology model is 'MCNiP'.

**Authors' response:** Accepted and appropriately changed;

**Text changed:** . . . . Drake et al. (2013) developed the Microbial Carbon and Nitrogen Physiology general model (MCNiP, Davidson et al. 2014) to estimate C and N rhizosphere cycling.

**Reviewer Report #5 (Referee #4):**

**Comment from Referees:** The works are mainly based on the data found in Mello et al. (1998), Neves (2000), Leles (2001), Teixeira et al. (2002), Gatto et al. (2003) and Maquere (2008). More similar works could be adopted in the work.

**Authors' response:** Thanks for the comment, but it is not clear that increasing the number of works would significantly improve the manuscript, as these six are already a representative of the forest plantation industry. Ideally a meta-analysis of all similar literature would be undertaken, but we feel it is unlikely to substantially change the model. Also, the referee has not provided any additionaly references on which could improve this aspect of the model, and such data are rare in the forestry literature. Therefore, at this stage, we consider that six references for the root model are sufficient, considering the scarcity of this information in the literature (fine root mass).

**Comment from Referees:** The English must be improved before resubmission or the manuscript can be checked or edited by two native English speakers with similar scientific backgrounds/of the same field. As for example, in the introduction, Page 2, Lines 7, 8, 13, 14, 16 etc.

**Authors' response:** The manuscript was rechecked and edited to improve English expression by one of our co-authors (P. Smethurst) whose first langauage is English.

**Comment from Referees:** The sources of the data used for modeling are not written in a recognized format.

**Authors' response:** We now indicate the data sources through numbering in the tables of the supplementary material.

**Comment from Referees:** Conclusions Page 19 line 4: the word should be "Conclusions". -The authors could not come up with the message of the work. Even, the sentences are not clear in expressing any meaning. All the sentences should be rewritten.

**Authors' response:** The conclusion section has been entirely rewritten, which also addresses a similar comment by the Associate Editor.

**Comment from Referees:** Minor concerns: Page 1, Line 19, 'and' should be used after the penultimate process.

**Text changed:** (...) and SOM formation.

**Comment from Referees:** 2.1 Parameter estimation Page 8, Lines 7-8 are not clear.

**Text changed:** (...) to estimate coefficients of the population dynamics modifiers due to the effect of water, soil organic matter and physical conditions.

**Comment from Referees:** 2.1.2 C and N availability and microbial demand Page 4, Lines 27-28: Sentence meaning should be clearer.

**Text changed:** The greater capacity of protection by the soil matrix prevents the microbiota to satisfy its nutritional demand, limiting the growth of the microbial population.

**Comment from Referees:** Page 6, Line 3: What does it mean by Eucalypt plantation root? Does it mean Eucalyptus plantation roots.

**Text changed:** The model simulates the effect eucalypts roots on the C and N cycling in the rhizosphere soil, (...)

**Comment from Referees:** 2.2 The evaluation of the rhizospheric model Page 8, Line 18: It is not clear what the authors wanted to say.

**Text changed:** A linear model of the type $\hat{y} = \boldsymbol{\beta}1x + \boldsymbol{\beta}0$ was adjusted, being the value predicted (P) by the model positioned on the X axis, and the value observed (O) in field experiments on the Y axis (literature data). The model performance was evaluated through the coefficient of determination ($R^2$). In addition, it was tested the coefficients $\boldsymbol{\beta}1$ considering the null hypothesis equal to 1; and the coefficient $\boldsymbol{\beta}0$ considering the null hypothesis equal to 0. An ideal model must have $\boldsymbol{\beta}1$ statistically not different from 1 and $\boldsymbol{\beta}0$ not different from 0. The t-test was used to evaluate these hypotheses.

**Comment from Referees:** Page 11, Line 30, 32, 35: Check. Sentences are not clear in meaning.

**Text changed:** In the case of a rhizosphere with a thickness of 0.5 mm, the volume of forest soil can reach 135,937.5 dm³, or approximately 5.4 % of the forest soil.

**Comment from Referees:** Page 13, Line 41; Sentence meaning is not clear.

**Text changed:** Table 3 shows the simulation of scenarios with soils with two carbon protection capacities (15 and 30 %).

**Comment from Referees:** Page 14, 6: the equivalent or 'equivalent'.

**Text changed:** equal to 24.6 % of N demand by the ecosystem (tree + litter) or 38.4 % of tree demand, which assumed losses of 40 % due to leaching, denitrification and volatilization

**Comment from Referees:** 2.1 Sensitivity analysis of the ForPRAN model: Lines 36-37 are not correct in meaning.

**Text changed:** The ranges of variables values were selected based on the natural variability found in nature.

**Comment from Referees:** Figure 8 and 9: Source: Based on not based in - In, table 1, soil clay contente should be "Clay content in soil".

**Text changed:** Based on Brock and Madigan (1991)./ Based on Silva et al. (2011)./ Based on Sato, Tsuyuzaki e Seto (2000)./ Clay content in soil (%)

**Comment from Referees:** Page 21, Line 2, Line 31, the meaning is not clear. Eq 19, 20, 27, 28 are not well-presented. Page 27, Lines 24-25, the sentences are not clear in meaning.

**Text changed:**

Page 21, Line 2: It was developed a multivariate empirical model being the dependent variable was the partitioning of the dry matter mass for fine roots (<= 3 mm) and, as independent variables, the soil clay content, soil layer thickness, and tree shoot mass.

Page 21, Line 31: We used the specific root length presented by Mello et al. (1998) and equations 4 and 5 to calculate specific root length (SRL, cm/g) of the specific root diameter class (SRLcd, cm/g) (equation 5).

Equations improved:

$$Jn = \begin{cases} 0, & \text{if is limited by C} \\ \left(Uc - Rm - \frac{EPc}{SUE}\right)\left(\frac{SUE}{CN_m}\right) - EPn - Un, & \text{if is limited by N} \end{cases} \qquad \text{Eq. 19}$$

$$Mn = \begin{cases} Un - EPn - \left(Uc - Rm - \frac{EPc}{SUE}\right)\left(\frac{SUE}{CN_m}\right), & \text{if is limited by C} \\ 0, & \text{if is limited by N} \end{cases} \qquad \text{Eq. 20}$$

$$Rg = \begin{cases} \left(Uc - \frac{EPc}{SUE} - Rm\right)(1 - SUE), & \text{if limited by C} \\ (Un - J_n - EP_n)CN_m \frac{(1-SUE)}{SUE}, & \text{if limited by N} \end{cases} \qquad \text{Eq. 27}$$

$$Ro = \begin{cases} 0, & \text{if limited by C} \\ \left(Uc - Rm - \frac{EP_c}{SUE}\right) - (Un + Jn - EPn)\frac{CN_m}{SUE}, & \text{if limited by N} \end{cases} \qquad \text{Eq. 28}$$

[revised manuscript text omitted]

---

## Author Response (AR2)

Viçosa, June 1$^{st}$, 2018

Dear Dr. Jens-Arne Subke

Thank you for your suggestions of 25$^{th}$ May 2018 for improving the article entitled "**Modeling rhizosphere carbon and nitrogen cycling in *Eucalyptus* plantation soil**" (Manuscript bg-2017-302), authored by Rafael V. Valadares, Júlio C. L. Neves, Maurício D. Costa, Philip J. Smethurst, Luiz A. Peternelli, Guilherme L. de Jesus, Reinaldo B. Cantarutti, and Ivo R. da Silva, to be considered for publication in the journal **Biogeosciences**.

The set of answers below show how the suggested changes were implemented. In addition, we will be pleased to address any further suggestions that you might have prior to acceptance.

Once again, we are very thankful for your editorial work and the improvements that it has prompted. Additionally, the manuscript has not been submitted for publication elsewhere and accordingly all authors contributed to the work described here and take entire responsibility for it. There is also no conflict of interest.

Sincerely yours,

Rafael V. Valadares
Departamento de Solos
Universidade Federal de Viçosa
36570-900 Viçosa, MG, Brazil
E-mail: rafaelvvaladares@hotmail.com
Tel:  +55 (31) 99253 7720

**Response to Associate Editor's Comments**

1- **Associate Editor's Comment:** P. 3, l. 8: "Eucalyptus rhizosphere cycling" is too vague. Make it clearer what you are aiming to model (i.e. C cycling only, or C&N combined?)

1- **Authors' response:** Accepted and appropriately modified.

1- **Text changed:** "propose a model for estimating rhizosphere C and N cycling in *Eucalyptus* plantation soil."

2- **Associate Editor's Comment:** p. 3, l. 19: No hyphen after "involves".

2- **Authors' response:** Accepted and appropriately modified.

2- **Text changed:** "This symbiosis involves *Eucalyptus* shoots and roots, soil microbes, and other soil properties . . .,"

3- **Associate Editor's Comment:** p. 3, l. 19: This is very Eucalypt-specific. Do you envisage this model to be applicable to other species (based on appropriate parameterisation), or is it specific to Eucalypts only? In that case, should the name of the tale reflect this?

3- **Authors' response:** Accepted and appropriately modified.

3- **Modified Text:** p.3, l. 18-21: This symbiosis involves shoots, roots, soil microbes, and other soil properties, the biological components of which may have co-evolved to sustain N and energy fluxes in the forest ecosystem. The application presented is for *Eucalyptus*, but the principles and model could be adapted to other plant-soil systems where data are available to guide parameterization. The process is schematically summarized in Figure 1.

4- **Associate Editor's Comment:** p. 5, l. 16-33: Something seems to be missing here. The section aims to outline modelling of microbial turnover, and you provide an overview of what is known about moisture and temperature effects. However, you don't indicate how this is then incorporated into ForPRAN. From the presentation, I would assume that moisture and temperature affect soil porosity, which is how microbial dynamics are implemented in eth model. Is that correct?

4- **Authors' response:** Accepted and appropriately modified.

4- **Text changed:** p.5, l. 24-26: "This effect is presented in ForPRAN by means of a modifier (Ku) in the microbial death rate (Kmf), the value of which is inversely proportional to water availability."

p. 5, l. 31-32; p. 6, l. 1-2: "We assumed that temperature influences enzymatic kinetics by being optimal in the range 25°C to 40°C and decreasing rapidly at higher and lower values. This effect was implemented in the ForPRAN model through the KappaD variable that influences the rate of SOM enzymatic depolymerization and, consequently, the rate of microbial growth."

p. 6, l. 6-7: "This effect is presented in ForPRAN by means of a modifier (Kpt) of the microbial death rate (Kmf), for which extreme values raise the Kmf rate in accordance with data present by Silva et al. (2011)."

5- **Associate Editor's Comment:** p. 8, l. 23-27: I propose a simplification of the text: "A linear model of the type $O = \beta_1 P + \beta_0$ was fitted, where P is the value predicted by the model, and O is the value observed in field experiments. Model performance was evaluated through the coefficient of determination ($R^2$). In addition, coefficient $\beta_1$ was tested for significant difference from 1, and coefficient $\beta_0$ for significant difference from 0 using t-tests.

5- **Authors' response:** Accepted and appropriately modified.

5- **Text changed:** p. 8, l. 23-26: " 1-A linear model of the type $O = \beta_1 P + \beta_0$ was fitted, where P is the value predicted by the model, and O is the value observed in field experiments. Model performance was evaluated through the coefficient of determination ($R^2$). In addition, coefficient $\beta_1$ was tested for significant difference from 1, and coefficient $\beta_0$ for significant difference from 0 using t-tests."

6- **Associate Editor's Comment:** p. 9, l. 20/21 (Figure caption): Simplify to: Correlation of fine root biomass reported in literature against values predicted using the ForPRAN model. The dotted line is the mean regression, with neither intercept nor slope significantly different from 0 or 1, respectively. Solid line: 1:1 relationship.

6- **Authors' response:** Accepted and appropriately modified.

6- **Text changed:** p. 9, l. 23-25: Regression of fine root biomass reported in literature against values predicted using the ForPRAN model. The dotted line is the mean regression, with neither intercept nor slope significantly different from 0 or 1, respectively. Solid line: 1:1 relationship.

7- **Comment from the Associate Editor:** p. 10, l. 16/17 (Caption of Fig. 4): Change to: Regression of microbial biomass observed by Drake et al. (2013) against results. Regression parameters were not significantly different from 1 (slope) and 0 (intercept), respectively." [explain insert table in graph]

7- **Authors' response:** Accepted and appropriately modified.

7- **Text changed:** p.10, l. 19-20: Regression of microbial biomass observed by Drake et al. (2013) against results. Regression parameters were not significantly different from 1 (slope) and 0 (intercept), respectively. Nash Sutcliffe Efficiency (NSE), Mean Error (ME), Mean Absolute Error (MAE), and Root Mean Square Error to Standard Deviation Ratio (RSR) are indicators of model efficiency and bias.

8- **Comment from the Associate Editor:** Pages 17 and 18: The figure captions are too short. These are modelled responses, right? State this clearly in the caption, so the reader can obtain all information necessary to understand the figure without reading the text! Please include the sol depth in the caption of Figure 10.

8- **Authors' response:** Accepted and appropriately modified.

8- **Text changed:** p. 17, l. 4: "Figure 7. Simulated temperature effect by the ForPRAN model on the kinetic enzyme variable KappaD. (Based on theoretical representation of Brock and Madigan, 1991)."

p. 17, l. 8-10: "Figure 8. Simulated effect of soil porosity by the ForPRAN model on the modifier of microbial death rate due to limitations of physical conditions (Kpt). (Source: The equation used in the Kpt modifier was parameterized with data presented in Silva et al., 2011)."

p. 18, l. 2-5: "Figure 9. Simulated effect of soil moisture by the ForPRAN model on the modifier of microbial death rate due to water limitation (Ku).

(Source: The equation used in the Ku modifier was parameterized with data presented in Sato et al., 2000).

p. 18, l. 8-10:"Figure 10. Boxplot of 206 observations of microbial biomass of soils under *Eucalyptus* growing in southeast Brazil (0-10 cm depth) (M. R. Tótola pers. comm.)"

9- **Comment from the Associate Editor:** P. 19, l. 5-8: This sentence is ambiguous. I think you're saying that you combine two models, one designed to simulate rhizosphere C and N dynamics, and another that simulates wood production. However, it reads more as if you have expanded an existing C/N model and now have a new product that simulates wood production. Please rephrase.

9- **Authors' response:** Accepted and appropriately modified.

[revised manuscript text omitted]

---

## Author Response (AR3)

Viçosa, June 26$^{th}$, 2018

Dear Dr. Jens-Arne Subke

Thank you for your suggestions of 12$^{th}$ June 2018 for improving the article entitled "**Modeling rhizosphere carbon and nitrogen cycling in *Eucalyptus* plantation soil**" (Manuscript bg-2017-302), authored by Rafael V. Valadares, Júlio C. L. Neves, Maurício D. Costa, Philip J. Smethurst, Luiz A. Peternelli, Guilherme L. de Jesus, Reinaldo B. Cantarutti, and Ivo R. da Silva, to be considered for publication in the journal **Biogeosciences**.

The answer below shows how the suggested changes was implemented. Additionally, the manuscript has not been submitted for publication elsewhere and accordingly all authors contributed to the work described here and take entire responsibility for it. There is also no conflict of interest.

Sincerely yours,

Rafael V. Valadares
Departamento de Solos
Universidade Federal de Viçosa
36570-900 Viçosa, MG, Brazil
E-mail: rafaelvvaladares@hotmail.com
Tel:  +55 (31) 99253 7720

**Response to Associate Editor's Comments**

1- **Associate Editor's Comment:** Table 3 is not very clear, I don't know what the numbers 15 and 30 in the first line indicate. It also isn't referred to in the text, which is not appropriate. Is this table necessary? If so, make sure it is well integrated to the text, and that all parts of it are clear.

1- **Authors' response:** Accepted and appropriately modified.

[revised manuscript text omitted]

---

## Author Response (AR4)

Viçosa, July 4th, 2018

Dear Dr. Jens-Arne Subke,

We are very grateful for all appropriate and constructive suggestions proposed to improve our article made through your editorial work and reviewer collaboration. All the authors received with great enthusiasm and fulfillment the final acceptance of the article entitled "**Modeling rhizosphere carbon and nitrogen cycling in *Eucalyptus* plantation soil**" (Manuscript bg-2017-302) for publication in the journal **Biogeosciences**.

We certificate that the manuscript has not been submitted for publication elsewhere and accordingly all authors contributed to the work described here and take entire responsibility for it. There is also no conflict of interest.

Sincerely yours,

Rafael V. Valadares
Departamento de Microbiologia
Universidade Federal de Viçosa
36570-900 Viçosa, MG, Brazil
E-mail: rafaelvvaladares@hotmail.com
Tel: +55 (31) 99253 7720